# Disentangled Representation Learning through Unsupervised Symmetry Group Discovery

**Barthélémy Dang-Nhu, Louis Annabi & Sylvain Argentieri**
Sorbonne Université, CNRS,
Institut des Systèmes Intelligents et de Robotique, ISIR,
F-75005 Paris, France
{dangnhu,annabi,argentieri}@isir.upmc.fr

## Abstract

Symmetry-based disentangled representation learning leverages the group structure of environment transformations to uncover the latent factors of variation. Prior approaches to symmetry-based disentanglement have required strong prior knowledge of the symmetry group's structure, or restrictive assumptions about the subgroup properties. In this work, we remove these constraints by proposing a method whereby an embodied agent autonomously discovers the group structure of its action space through unsupervised interaction with the environment. We prove the identifiability of the true symmetry group decomposition under minimal assumptions, and derive two algorithms: one for discovering the group decomposition from interaction data, and another for learning Linear Symmetry-Based Disentangled (LSBD) representations without assuming specific subgroup properties. Our method is validated on three environments exhibiting different group decompositions, where it outperforms existing LSBD approaches.

## 1 Introduction

An important property of a representation is its disentanglement, as it enables a form of interpretability (Higgins et al., 2017), fairness (Locatello et al., 2019a), improved transferability (Lee et al., 2021; Bengio et al., 2020), and the ability to directly manipulate the latent space (Kim & Mnih, 2018; Chen et al., 2016). For this reason, many unsupervised disentangled representation learning methods have been proposed, initially relying on Variational Autoencoders (VAEs) (Kim & Mnih, 2018; Kumar et al., 2018; Higgins et al., 2017) or Generative Adversarial Networks (GANs) (Chen et al., 2016). Locatello et al. (2019b) showed that unsupervised disentanglement requires additional prior knowledge or inductive biases. Thus, several approaches, relying on different additional assumptions, address the question of unsupervised disentangled representation learning.

Learning disentangled representations relies on the assumption that there exist true underlying factors of variation in the environment (Bengio et al., 2013), and aims to infer them from available observations. The symmetry-based approach of Higgins et al. (2018) proposes to achieve such disentanglement by exploiting the subgroup decomposition of the group of environment transformations, called symmetries. Each subgroup is associated with a specific part of the representation. When a symmetry from a particular subgroup is applied, only the corresponding part of the representation varies. Caselles-Dupré et al. (2019) demonstrated that symmetry-based disentanglement is only possible when access is granted to transitions (initial observation, transformation, resulting observation). Several notable works follow this approach (Quessard et al., 2020; Tonnaer et al., 2022; Keurti et al., 2023). However, they rely on restrictive assumptions regarding the nature of the symmetry group or prior knowledge about its structure. This paper aims to overcome these limitations by designing algorithms and providing proofs for the autonomous discovery of the symmetry group structure, and its exploitation for disentangled representation learning.

Our contributions are as follows:

- We prove, under certain assumptions, the identifiability of the ground-truth group decomposition of the symmetry group from a dataset of transitions.

- We derive from this theorem an algorithm for the discovery of the symmetry group decomposition.

- We introduce a novel method for learning a LSBD representation directly from a group decomposition, without imposing any structural assumptions on the subgroups, and we provide theoretical guarantees of disentanglement under specific assumptions.

- We combine these two algorithms and show experimentally that the full method outperforms other LSBD methods on several datasets with different group structures.

## 2 PRELIMINARIES

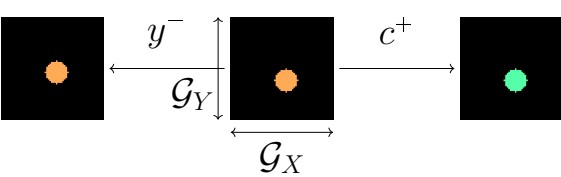

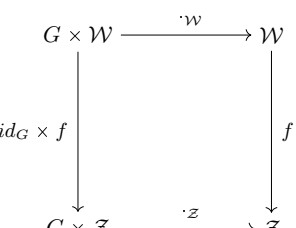

Figure 1: Colored Flatland environment. The group of symmetries can be decomposed as $G = G_x \times G_y \times G_c$ corresponding respectively to the cyclic groups of translations on the horizontal axis/vertical axis, and in a list of predefined colors. The agent has access to several symmetries (or actions) $\mathcal{G}_x = \{x^+, x^-\} \subset G_x$, $\mathcal{G}_y = \{y^+, y^-\} \subset G_y$, and $\mathcal{G}_c = \{c^+, c^-\} \subset G_c$.

Figure 2: Equivariance property.

We consider the framework of Linear Symmetry-Based Disentanglement (LSBD) (Higgins et al., 2018), which provides a formal definition of disentanglement suitable for deriving identifiability results and guiding the design of representation learning algorithms. Let $\mathcal{W}$ denote the set of possible environmental states. We define a generative process $b : \mathcal{W} \to \mathcal{X}$ that maps a state to an observation, and an encoder $h : \mathcal{X} \to \mathcal{Z}$ that maps observations into a latent representation. The overall mapping is then given by $f = h \circ b : \mathcal{W} \to \mathcal{Z}$. We assume that $b$ is an intrinsic (and unknown) property of the environment, while $h$ is agent-specific and can be learned.

We further assume the existence of a symmetry group $G$ acting on $\mathcal{W}$. A key assumption is that $G$ satisfies the standard group axioms: the existence of an identity element, closure under composition, and the existence of inverses. This group structure enables the definition of a group action $\cdot_{\mathcal{W}} : G \times \mathcal{W} \to \mathcal{W}$, which maps each pair $(g, w) \in G \times \mathcal{W}$ to a transformed world state $w' \in \mathcal{W}$ resulting from the application of $g$. The agent is endowed with an action set $\mathcal{G} \subset G$ that contains only a subset of the full group. Crucially, $\mathcal{G}$ is not required to form a group itself, in particular, the agent's actions may not be reversible, and the identity element of $G$ may not be included in $\mathcal{G}$.

We also assume that the group $G$ admits a decomposition into a direct product of subgroups, i.e., $G = G_1 \times \cdots \times G_K$. For example, in the Flatland environment illustrated in Figure 1, the symmetry group $G$ can be decomposed into three subgroups corresponding to cyclic groups of horizontal translations, vertical translations, and color shifts.

**Definition 1** (Linear Symmetry Based Disentanglement). *A representation $h$ is said to be symmetry-based disentangled (SBD) with respect to $\langle \mathcal{W}, b, \prod_k G_k \rangle$ if:*

1. *There exists a group action $\cdot_{\mathcal{Z}} : G \times \mathcal{Z} \to \mathcal{Z}$,*

2. *Equivariance holds: $\forall g \in G, w \in \mathcal{W}$, we have $g \cdot_{\mathcal{Z}} f(w) = f(g \cdot_{\mathcal{W}} w)$,*

3. *There exists a decomposition $\mathcal{Z} = \mathcal{Z}_1 \oplus \cdots \oplus \mathcal{Z}_K$ and group actions $\cdot_k : G_k \times Z_k \to Z_k$ such that*
$$(g_1, \ldots, g_K) \cdot_{\mathcal{Z}} (z_1, \ldots, z_K) = (g_1 \cdot_1 z_1, \ldots, g_K \cdot_K z_K),$$

*4. The function $h$ is injective*

*Moreover, the representation is said to be linearly disentangled (LSBD) if $\cdot_\mathcal{Z}$ is linear, i.e. there exists a representation $\rho : G \to GL(\mathcal{Z})$ such that $g \cdot_\mathcal{Z} z = \rho(g)z$.*

The original definition provided by Higgins et al. (2018) does not explicitly state the fourth condition requiring the encoder $h$ to be injective. However, this constraint is implicitly assumed within the LSBD framework; without it, any constant mapping would trivially satisfy the LSBD criteria. Caselles-Dupré et al. (2019, Theorem 1) demonstrated that learning an LSBD representation is impossible from observations alone, they proved that multiple distinct world sets and group actions can produce the same set of observations. However, these environments differ in their transitions. Consequently, they proposed leveraging transitions of the form $(x, g, x')$ rather than relying solely on passive observations $x$. This perspective naturally aligns with the reinforcement learning setting, where agents can actively interact with the environment by performing actions that induce state transitions. Accordingly, in the remainder of this work, we refer to symmetries $g$ as actions. Mathematical background and a symbol table can be found in Appendix A.

## 3 RELATED WORK

Several methods have been proposed to learn LSBD representations from transitions, all relying on auto-encoder architectures. *Forward-VAE* (Caselles-Dupré et al., 2019) augments the evidence lower bound (ELBO) of a VAE with a latent-space action loss. Disentanglement is encouraged by constraining the matrices $\rho(g)$ to follow a predefined structure, which requires prior knowledge of the subgroup decomposition of the symmetry group, as well as the minimal number of latent dimensions assigned to each subgroup. Another method, proposed by Quessard et al. (2020), referred to as *SO-Based Disentangled Representation Learning* (SOBDRL), aims to learn representations with a prediction loss that aim to infer the next observation $x'$ from $(x, g)$. The action matrices are parameterized as elements of the special orthogonal group $SO(d)$, the disentanglement is encouraged with a regularization term that minimizes the number of latent dimensions involved in each transformation, encouraging transformations constrained to $SO(2)$. *LSBD-VAE*, introduced by Tonnaer et al. (2022), relies on the $\Delta$-VAE architecture (Rey et al., 2019), which supports latent spaces defined over arbitrary manifolds. In this framework, both the group decomposition $G = G_1 \times \cdots \times G_K$ and its representation $\rho$ are assumed to be known a priori. This prior knowledge allows the model to align the latent geometry with the group structure and to incorporate an action-aligned loss term, in the spirit of Forward-VAE. *Homomorphism AutoEncoder* (HAE), proposed by Keurti et al. (2023), assumes that $G$ is a Lie group and that the agent has access to $\varphi(g)$, where $\varphi$ is an unknown nonlinear mapping. The action representation is learned from this mapping by jointly predicting both current and future states in the observation and latent spaces. Disentanglement is encouraged by enforcing a block-diagonal structure on the action matrices.

Some other methods aim to learn LSBD representation solely from the observation based on the metric of the ground-truth Riemannian manifold (Pfau et al., 2020), the cardinality of each cyclic group (Yang et al., 2022) or the commutativity of the whole symmetry group (Zhu et al., 2021).

We observe that all state-of-the-art methods based on transitions rely on assumptions regarding the structure of the symmetry group or its subgroups. In contrast, the goal of this work is to relax these assumptions by introducing a symmetry-based disentangled representation learning approach that does not require any prior knowledge of the group decomposition.

## 4 METHODS

We suppose that the available actions $\mathcal{G}$ are a subset of the whole group action $G$ and that there is a dataset $\mathcal{D}$ of transitions $(x, g, x')$ where $g \in \mathcal{G}$ are the indices of the actions taken by the agent. We aim to learn an LSBD representation $h$. Our method consists of three steps:

1. We learn an entangled representation *i.e.* a representation satisfying only points 1, 2 and 4 of Definition 1 to learn an action representation $\rho : G \to GL(\mathcal{Z})$ and an encoder $h : \mathcal{X} \to \mathcal{Z}$.

2. From $\rho$ and $h$ we compute a decomposition $G = G_1 \times \cdots \times G_K$ by regrouping actions using a custom pseudo-distance based on group theory.

3. From this decomposition we learn a disentangled representation.

## 4.1 (STEP 1) LEARN AN ENTANGLED REPRESENTATION

Our objective is to learn an encoder $h : \mathcal{X} \to \mathcal{Z} = \mathbb{R}^d$ and an action representation $\rho_\psi : \mathcal{G} \to \mathbb{R}^{d \times d}$ satisfying the equivariance property defined in Definition 1. As there is no prior knowledge about th action matrices, each matrix $\rho_\psi(g)$ is directly parameterized by $d^2$ learnable scalars, resulting in a total of $|\mathcal{G}| \times d^2$ parameters. To perform this step, we introduce a method referred to as *Action-based VAE* (A-VAE), which builds upon the variational autoencoder (VAE) framework (Kingma & Welling, 2014; Rezende et al., 2014). The goal is to map each observation $x \in \mathcal{X}$ to a latent representation in $\mathcal{Z} = \mathbb{R}^d$. Let $\tau = (x, g, x')$ denote a transition, and let $z$ and $z'$ be the corresponding latent representations of $x$ and $x'$, respectively. The model architecture is illustrated in Figure 3 and defined as follows:

- $p_\theta(X'|z') = \mathcal{N}\big(\mu_\theta(z'), Diag(\sigma_\theta(z')^2)\big),$    (1)
- $p_{\psi,\phi}(Z'|x,g) = \mathcal{N}\big(\rho_\psi(g)\mu_\phi(x), Id\big),$    (2)
- $q_\phi(Z'|x') = \mathcal{N}\big(\mu_\phi(x'), Diag(\sigma_\phi(x')^2)\big).$    (3)

Figure 3: Graphical model

In contrast to the standard VAE, we condition the prior distribution over $Z'$ on both the past observation $x$ and the action $g$. To maximize the expected log-likelihood $\mathbb{E}[\log p(\tau)]$ with respect to the model parameters $\theta$, $\phi$, and $\psi$, we derive the corresponding evidence lower bound (ELBO) for our graphical model. As shown in Appendix B, we obtain (up to an additive constant):

$$\log p(\tau) \geqslant - \frac{1}{2} \left\| \rho_\psi(g)\mu_\phi(x) - \mu_\phi(x') \right\|^2 - \frac{1}{2}\|\sigma_\phi(x')\|^2 + \sum_i \log \sigma_\phi(x')_i \quad \Big\} \text{ action part}$$

$$- \mathbb{E}_{z' \sim q_\phi(z'|x')} \left[ \sum_i \log \sigma_\theta(z')_i + \left\| \frac{x' - \mu_\theta(z')}{\sigma_\theta(z')} \right\|^2 \right] \quad \Big\} \text{ reconstruction part}$$

$$(4)$$

Analogously to $\beta$-VAE (Higgins et al., 2017), we introduce a weighting coefficient to balance the two components of the objective, resulting in the loss function $\mathcal{L} = \mathcal{L}_{\text{REC}} + \lambda_{\text{ACT}}\mathcal{L}_{\text{ACT}}$. Each of the three conditional distributions in the model is implemented using deep neural networks trained via backpropagation. The model parameters $\theta$, $\phi$, and $\psi$ are optimized to maximize the ELBO. As in standard VAEs, we apply the reparameterization trick to enable gradient-based optimization through the reconstruction term. In practice, the standard deviations $\sigma_\theta$ and $\sigma_\phi$ are fixed. Details of the neural network architectures are provided in Appendix I.2.

## 4.2 (STEP 2) LEARN THE GROUP STRUCTURE

Once the action representation $\rho_\psi$ and the encoder $h = \mu_\theta$ have been learned, we aim to leverage them to recover the group decomposition $G = G_1 \times \cdots \times G_K$. By abuse of notation, we will treat the direct factors $G_i$ as subgroups of $G$.

### 4.2.1 ASSUMPTIONS

**Assumption 1.** *The environment is fully observable i.e. the observation function $b : \mathcal{W} \to \mathcal{X}$ is injective.*

It is a strong assumption, however it is necessary as we have the following result:

**Theorem 1.** *For a SBD representation to exist, it is necessary for the observation function $b$ to be injective (up to an interaction equivalence class).*

The definition of the *interaction equivalence class* and the proof of the theorem are provided in Appendix G. The key idea is that components of the world state that do not influence the agent's

interaction can be discarded, yielding an equivalent environment from the agent's perspective. In this reduced environment, the observation function must be injective for a SBD representation to exist. Although this assumption is not always stated explicitly, it is in fact a necessary condition for all SBD representation learning algorithms and is not specific to our method.

The next two assumptions are assumptions specific to the proposed algorithm, and are intended to replace the stronger prior assumption commonly made in the SBD literature consisting in providing prior knowledge of the group decomposition. We first assume that each action belongs to a unique subgroup $G_i$. We refer to this property as disentanglement of the action set with respect to $\prod_k G_k$. It is a strong assumption but we demonstrate empirically in Appendix H.2 that related SBD methods make a similar implicit assumption.

**Assumption 2.** *$\mathcal{G}$ is disentangled with respect to $G = \prod_k G_k$. That is, $\mathcal{G} = \mathcal{G}_1 \cup \cdots \cup \mathcal{G}_K$ with $\forall k, \mathcal{G}_k \subset G_k$.*

We argue that this assumption alone is not sufficient to recover the correct decomposition. To illustrate this point, consider two distinct environments analogous to Flatland shown Figure 4: (a) a $2 \times 3$ cyclic grid *i.e.* $G^a = \mathbb{Z}/2\mathbb{Z} \times \mathbb{Z}/3\mathbb{Z}$ with actions $\mathcal{G}^a = \{x^+\} \cup \{y^+\}$ and (b) a $6 \times 1$ cyclic grid *i.e.* $G^b = \mathbb{Z}/6\mathbb{Z}$ with actions $\mathcal{G}^b = \{2x^+, 3x^+\}$. Both environments satisfy Assumption 2 and can share the same representation, as there exists an isomorphism from $G^a$ to $G^b$ that maps each element of $\mathcal{G}^a$ to a corresponding element in $\mathcal{G}^b$. From the agent's perspective, these two situations are indistinguishable in the absence of additional assumptions. Ideally, we seek an assumption that both *covers* a wide range of practical scenarios, i.e. action sets $\mathcal{G}$, and enables a *computationally tractable* procedure for recovering the group

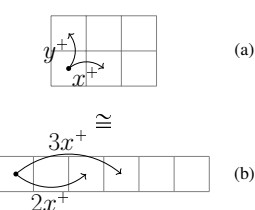

Figure 4: Two isomorphic group actions satisfying Assumption 2.

decomposition. Among the various options considered, we adopt the following assumption, as it offers a favorable trade-off between situation coverage and computational feasibility:

**Assumption 3.** *For all $g, g' \in \mathcal{G}$ with $g \neq g'$, if they belong to the same subgroup then there exists $u \in \mathcal{G}$ and $m \in [\![1, M]\!]$ such that we have either $g = u^m g'$, $g = g' u^m$, $g' = g u^m$ or $g' = u^m g$.*

Combined with Assumption 2, it is straightforward to show that the implication of Assumption 3 is in fact an equivalence. As a result, we obtain a simple and practical criterion for determining whether two actions belong to the same subgroup. In terms of situation coverage, as soon as $M \geq 2$, Assumption 3 holds in common cases such as when $\mathcal{G}_i$ contains an action and its inverse, when $\mathcal{G}_k = G_k$, or when $\mathcal{G}_k = G_k^*$. In practice, the action sets considered in the experimental sections of state-of-the-art SBDRL algorithms typically fall into one of these categories. In the scenario illustrated in Figure 4, Assumption 3 allows us to assume that situation (b) will never occur, our method will thus assume that the environment corresponds to case (a).

### 4.2.2 ALGORITHM

We now introduce a method to recover the group decomposition i.e. to cluster the available actions into subgroups. Given an encoding function $h : \mathcal{X} \to \mathbb{R}^d$ and a matrix $A \in \mathbb{R}^{d \times d}$, we define the following semi-norm:

$$\|A\|_h = \mathbb{E}_x \left[ \|Ah(x)\| \right]. \tag{5}$$

From this and Assumption 3, we define the following pseudo-distance to determine whether two actions belong to the same subgroup. We write $A_g$ instead of $\rho_\psi(g)$ for simplicity and readability:

$$d_G(g, g') = \min_{\substack{u \in \mathcal{G} \\ m \in [\![1, M]\!]}} \min \left\{ \|A_g - A_u^m A_{g'}\|_h; \|A_g - A_{g'} A_u^m\|_h; \|A_{g'} - A_u^m A_g\|_h; \|A_{g'} - A_g A_u^m\|_h \right\}. \tag{6}$$

**Theorem 2.** *If the Assumption 1 to 3 are satisfied, the dataset contains all the possible transitions, $\mathcal{W}$ is finite and the A-VAE loss converges toward its minimum, then at some point of the training, two available actions will belong to the same subgroup if and only if their distance with respect to $d_G$ is below a specific threshold $\eta$ computed from $h$ and $\rho_\psi$.*

Based on Theorem 2, we design a clustering algorithm that groups together actions $g$ and $g'$ whenever $d_G(g, g') \leqslant \eta$. The choice of the threshold $\eta$, the details of the algorithm, and the proof are provided in Appendix C.

Once the group decomposition has been recovered, a suitable disentangled representation learning algorithm can be applied. For example, if $\mathcal{G} = \{g_1, g_1^{-1}\} \cup \{g_2, g_2^{-1}\}$, then $G$ is isomorphic to a subgroup of $SO(2) \times SO(2)$, and Forward-VAE (Caselles-Dupré et al., 2019) can be employed with an appropriate parameterization. However, as discussed in Section 3, existing LSBD methods still rely on some form of prior knowledge about the group structure. In the following section, we address this limitation by introducing a new disentangled representation learning algorithm that does not require such prior information.

### 4.3 (STEP 3) LEARN A DISENTANGLED REPRESENTATION

Now that we have the symmetry group decomposition, we aim to find a linear disentangled representation *i.e.* a decomposition $\mathcal{Z} = \mathcal{Z}_1 \oplus \cdots \oplus \mathcal{Z}_K$ and an action representation $\rho = \rho_1 \oplus \cdots \oplus \rho_K$ such that for each action $g = (g_1, \ldots, g_K) \in G$ and latent factor $z = (z_1, \ldots, z_K)$ we have $\rho(g_1, \ldots, g_K)(z_1, \ldots, z_K) = (\rho_1(g_1)z_1, \ldots, \rho_K(g_K)z_K)$.

This definition allows the $\mathcal{Z}_i$ to be any sub-vector spaces of $\mathcal{Z}$ as long as they form a direct sum, they are not required to be orthogonal. In our method, we additionally choose to search for representations where the disentanglement aligns with Cartesian axes of the latent space $\mathcal{Z} = \mathcal{Z}_1 \times \cdots \times \mathcal{Z}_K$. This choice is motivated by the fact that, under most widely accepted definitions of disentanglement, each latent dimension is expected to encode information about at most one ground-truth factor of variation (Wang et al., 2024). Consequently $\rho(g_1, \ldots, g_K)(z_1, \ldots, z_K) = \rho(g)z$ with $z = concat(z_1, \ldots, z_K)$ and $\rho(g) = diag(\rho_1(g_1), \ldots, \rho_K(g_K))$.

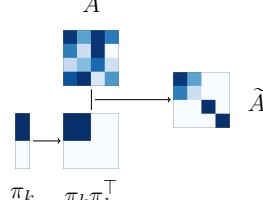

Thanks to Assumption 2, each action is known to belong to a unique subgroup. Consequently, for any $g \in G_k$ and $k' \neq k$, we have $\rho_{k'}(g)$ equal to the identity transformation. In matrix terms, this implies that each action is represented by the identity matrix, except for a single block along the diagonal, as illustrated in Figure 5 with the matrix $\widetilde{A}$ (in practice, the indices of the matrix may be permuted, however for the sake of clarity, we illustrate only the case in which the active dimensions are adjacent). Learning the structure of these matrices amounts to assigning each latent dimension $i$ to a unique subgroup $G_k$. Let $\pi_k \in \{0, 1\}^d$ denote the binary indicator vector encoding the set of dimensions assigned to the $k$-th subgroup, such that $\sum_k \pi_{k,i} = 1$.

Figure 5: Masking used to build disentangled action matrices

To enforce the desired block structure in the action matrices, we apply the mask $\pi_k \pi_k^\top$ to unstructured action matrices $A$ as illustrated in Figure 5. Let $k(g)$ denote the index of the subgroup to which the action $g$ belongs, and let $\odot$ denote the element-wise product. The structured action matrix $\widetilde{A}_g$ is then defined as:

$$\widetilde{A}_g = \pi_{k(g)} \pi_{k(g)}^\top \odot A_g + (1 - \pi_{k(g)} \pi_{k(g)}^\top) \odot I \tag{7}$$

To learn the vectors $\pi_k$, we employ a continuous relaxation. Specifically, we use $d$ softmax operations to ensure that $\pi_{k,i} \in [0, 1]$ with $\sum_k \pi_{k,i} = 1$. In order to promote disentanglement, we introduce an additional term in the A-VAE loss function that encourages the vectors $\pi_k$ to be close to be binary. A natural approach is to minimize the entropy $\mathcal{H}(\pi) = \sum_i \mathcal{H}(\pi_{:,i})$. However, empirical observations show that directly minimizing this entropy causes it to collapse to zero before the other loss components begin to decrease, leading to a random dimension assignment. To address this issue, we define the disentanglement loss as $\mathcal{L}_{DIS} = \sum_i |\mathcal{H}(\pi_{:,i}) - C|$, where $C$ is a target entropy value that is gradually annealed from its maximum to zero during training. We refer to this method as the *Group-Masked Action-based VAE* (GMA-VAE). The following result, proven in Appendix D, formalizes the disentanglement guarantee:

**Theorem 3.** *If Assumptions 1 and 2 are satisfied, the dataset contains all the transitions and $G$ is finite, then the encoders minimizing the GMA-VAE loss are LSBD representations with respect to $\langle \mathcal{W}, b, \prod_k \langle \mathcal{G}_k \rangle \rangle$ with $\langle \mathcal{G}_k \rangle$ representing the subgroup generated by $\mathcal{G}_k$.*

## 5 RESULTS

### 5.1 EXPERIMENTS

**Metrics**: To evaluate the disentanglement, we use the Independence (Inde) metric (Painter et al., 2020) that was specifically designed for the LSBD framework; we will also use classical disentanglement metrics: $\beta$-VAE (Higgins et al., 2017), Mutual Information Gap (MIG) (Chen et al., 2018), DCI disentanglement metric (Eastwood & Williams, 2018), Modularity (Mod) (Ridgeway & Mozer, 2018) and SAP (Kumar et al., 2018). All these metrics take values between 0 and 1 and are meant to be maximized.

**Algorithms**: We categorize the baseline methods into three classes: (1) *Supervised methods* where the action representation $\rho$ is given. The only supervised method is LSBD-VAE (Tonnaer et al., 2022). (2) *Self-supervised methods* where $\rho$ is learned as in SOBDRL (Quessard et al., 2020). We introduce a modified LSBD-VAE in which the action representation $\rho$ is learned rather than provided, we refer to this variant as method LSBD-VAE*. As also reported in Tonnaer et al. (2022), we were unable to obtain satisfactory results with Forward-VAE (Caselles-Dupré et al., 2019) on our datasets, therefore it is not included among the baselines. An other method is HAE (Keurti et al., 2023) that is specifically designed for Lie groups and therefore is only used in Section 5.6. (3) *Unsupervised methods* which rely solely on observations rather than transitions. This category includes classical disentanglement approaches: $\beta$-VAE (Higgins et al., 2017), Factor-VAE (Kim & Mnih, 2018) and DIP-VAE I/II (Kumar et al., 2018).

**Latent dimension**: For A-VAE we arbitrarily chose a latent dimension of 13, for the LSBD methods we chose the minimal dimension depending on the method and the symmetry group. Those minimal dimensions are discussed in Appendix E.

**Environments**: Similarly to *Flatland* (Caselles-Dupré et al., 2018), our first environment consists of a disk moving along the $x$ and $y$ axes over a black background as illustrated Figure 1. Additionaly to the groups acting on the position of the disk, a third group acts on the color feature and can be either a cyclic shift of the RGB channels, corresponding to $G_C = \mathbb{Z}/3\mathbb{Z}$ with $\mathcal{G}_C = \{c^-, c^+\}$, or a full permutation group over the RGB channels, i.e., $G_C = \mathfrak{S}_3$ with $\mathcal{G}_C = \mathfrak{S}_3^*$. The second environment is based on the *COIL dataset* (Nene et al., 1996), which contains images of objects captured from multiple viewpoints. Each observation consists of $n$ adjacent objects. Each object $i \in [\![1, n]\!]$ can be rotated through $k_i$ discrete angles, forming a cyclic rotation group $G_{R_i} = \mathbb{Z}/k_i\mathbb{Z}$, with the action set $\mathcal{G}_{R_i} = \{r_i^-, r_i^+\}$. In addition, the objects can be permuted via the symmetric group $G_{\mathfrak{S}} = \mathfrak{S}_n$. The third environment use the *3DShapes dataset* (Burgess & Kim, 2018), which consists of rendered images of a 3D object placed in a colored room. The data is generated from six discrete ground-truth factors: wall hue, object hue, background hue, object scale, object shape, and viewing angle. For each factor $i$, we define a cyclic symmetry group $G_i = \mathbb{Z}/k_i\mathbb{Z}$ and an action set consisting of two shifts, $\mathcal{G}_i = \{g_i^-, g_i^+\}$, corresponding to increments and decrements along the factor axis. The final dataset is *MPI3D* (Gondal et al., 2019), which consists of realistic images of a robotic arm moving an object. Among the various factors of variation available in the dataset, we retain only the horizontal and vertical rotation angles of the arm. Both are modeled as cyclic groups, $G_H = \mathbb{Z}/40\mathbb{Z}$ and $G_V = \mathbb{Z}/40\mathbb{Z}$, with corresponding action sets $\mathcal{G}_H = G_H^*$ and $\mathcal{G}_V = G_V^*$.

### 5.2 ACTION CLUSTERING

To evaluate the action clustering performance of Step 2, we use the Flatland environment with cyclic color shifts (FLC) and color permutations (FLP), as well as the COIL dataset with two (COIL2) and three (COIL3) objects. Our algorithm successfully recovers the ground-truth group decomposition in 100% of runs. The average group distances across random seeds are reported in Appendix H.5. In these experiments, the datasets include all possible transitions, and the available actions are simple (e.g., an action and its inverse). To assess the robustness of our method, we consider more challenging settings with both complex action sets and limited transition coverage. For this purpose, we use the COIL environment with three or four objects and random action sets $\mathcal{G}$ satisfying Assumptions 2 and 3, those environments are given Appendix I.1. In this setting, for each state $w \in \mathcal{W}$, we randomly sample $n_a \leqslant |\mathcal{G}|$ available actions to be used in the dataset. The results show that, as soon as $n_a \geqslant 2$, the method consistently recovers the correct group decomposition. Importantly, the same hyperparameters are used across all of these latter experiments.

## 5.3 DISENTANGLEMENT

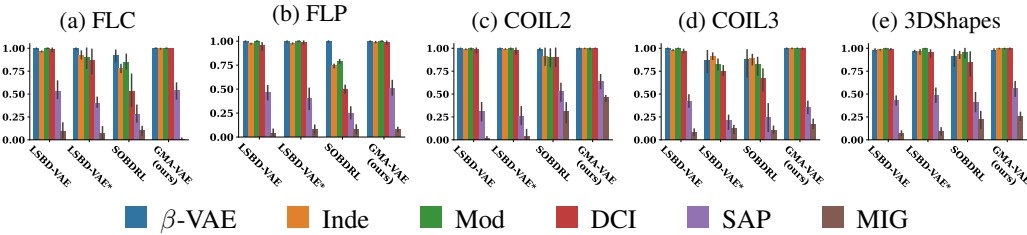

Figure 6: Median of disentanglement metrics. Vertical lines indicates the 25th to 75th percentile

To evaluate the disentanglement we use the same environments as before, the disentangled results are shown Figure 6. For more clarity we do not present the disentanglement of unsupervised methods as they perform significantly worse than LSBD methods. Detailed results including unsupervised methods are available in Appendix H.4.

The first observation is that all methods perform poorly in MIG and SAP as these two metrics require each ground truth factor of variation to be encoded in a unique dimension. However, linear disentanglement mostly requires features to be encoded in at least two dimensions as discussed in Appendix E. The only exception is COIL2 as the permutation group $\mathfrak{S}_2$ can be encoded in only one dimension with our method. The second observation is that our method performs almost perfectly for the other metrics and yields a disentanglement comparable to the supervised method LSBD-VAE.

## 5.4 LONG-TERM PREDICTION

We aim to investigate the effect of disentanglement on long-term prediction accuracy. To this end, we use the trained models to predict a final observation given an initial observation and a sequence of actions. For the COIL2 dataset, SOBDRL fails to consistently learn a disentangled representation. We therefore separate the seeds into two groups: those where disentanglement is achieved and those where it is not. On the COIL3 dataset, SOBDRL is unable to disentangle the representation at all, as the method is not suited to permutation-based symmetries. We also omit the results of LSBD-VAE* on COIL3, as it fails to consistently produce accurate predictions even for single-step transitions.

Figure 7 shows the prediction error as a function of the sequence length. We observe a drop in the prediction error of SOBDRL on COIL2. This behavior comes from the fact that disentangled actions of SOBDRL are $SO(2)$ rotations. In one of the seeds, the action that swaps the two objects has a larger angular error than the other actions, causing the corresponding latent dimensions to diverge first. However, due to the cyclic nature of $SO(2)$, the accumulated angular errors eventually cancel out, completing a full rotation and temporarily restoring the correct latent representation.

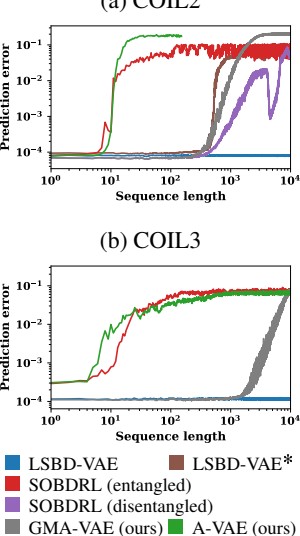

Figure 7: Median of long-term prediction error, the shaded area indicates the 25th to 75th percentile

Overall, three types of behavior emerge from the results. First, entangled self-supervised methods (A-VAE and SOBDRL) achieve good short-term predictions but quickly diverge as the sequence length increases. In particular, the A-VAE curve ends early because the latent representations eventually diverge to NaN values. Second, disentangled self-supervised methods (GMA-VAE, SOBDRL and LSBD-VAE*) achieve significantly better long-term predictions. Finally, the supervised method LSBD-VAE achieves perfect prediction performance regardless of sequence length. This is explained by the fact that, with access to ground-truth action matrices, the model satisfies exactly $A_g A_{g'} = A_{gg'}$, making multi-step prediction no more difficult than single-step prediction.

## 5.5 GENERALIZATION

To assess how disentanglement impacts generalization, we train each model on COIL2 and COIL3 using restricted datasets. We first consider the *independant and identicaly distributed* (iid) setting, in which the training and test sets follow the same distribution. Specifically, for each state, we uniformly sample $n_a = |\mathcal{G}|/2$ actions to include in the training data. The second experiment assesses the *out-of-distribution* (ood) generalization capabilities of the models. In this setting, the training set is restricted to transitions in which only the right-most object is allowed to rotate. We evaluate the prediction error on both seen and unseen transitions, the results are reported in Table 1 using the format seen / unseen prediction error. We highlight in bold the methods for which the error increases by less than 5% between seen and unseen transitions. In both experiments, we observe that all disentangled methods generalize well, while most entangled methods exhibit poor generalization, particularly in the *ood* setting.

Table 1: *iid* and *ood* prediction error, the format used is seen / unseen prediction error

| | | iid | | ood | |
| --- | --- | --- | --- | --- | --- |
| | | COIL2 | COIL3 | COIL2 | COIL3 |
| LSBD-VAE | | **7.8e-5 / 7.9e-5** | **1.1e-4 / 1.1e-4** | **7.6e-5 / 7.6e-5** | **9.9e-5 / 9.9e-5** |
| LSBD-VAE* | | **8.7e-5 / 8.8e-5** | | **8.8e-5 / 8.8e-5** | |
| SOBDRL | Disentangled | **1.7e-4 / 1.7e-4** | | **5.1e-5 / 5.1e-5** | |
| | Entangled | 1.7e-4 / 3.7e-3 | **2.5e-4 / 2.5e-4** | 5.7e-5 / 0.02 | 2.5e-4 / 0.01 |
| GMA-VAE | | **6.1e-5 / 6.2e-5** | **1.1e-4 / 1.1e-4** | **6.2e-5 / 6.2e-5** | **1.1e-4 / 1.1e-4** |
| A-VAE | | **7.7e-5 / 7.8e-5** | 2.9e-4 / 8.7e-4 | 6.7e-5 / 0.05 | 2.9e-4 / 0.05 |

## 5.6 LIE GROUPS

Here, we aim to investigate how our proposed method can be extended to continuous groups, such as Lie groups. The action clustering procedure of Step 2 cannot be applied to continuous groups since it relies on a finite clustering process. Nevertheless, when the group decomposition is assumed to be known, GMA-VAE can still be used to learn a disentangled representation, as the proof of Theorem 3 can be adapted to continuous symmetry groups:

**Theorem 3'.** *If Assumptions 1 and 2 hold, the dataset contains all transitions, and the monoid generated by each $\mathcal{G}_k$ is $G_k$, then the encoders that minimize the GMA-VAE loss are LSBD representations with respect to $\langle \mathcal{W}, b, \prod_k G_k \rangle$.*

To empirically validate this claim, we consider the MPI3D dataset, in which actions correspond to continuous rotations of a robotic arm. The symmetry group is modeled by $SO(2) \times SO(2)$, each subgroup corresponding to an axis of rotation of the arm. The action representation $\rho : g \mapsto \rho(g)$ is implemented as a neural network, while the rest of the GMA-VAE architecture remains unchanged. As reported in Figure 8a, GMA-VAE achieves performance comparable to SOBDRL and outperforms HAE.

To evaluate the robustness of GMA-VAE to noise, we introduce Gaussian noise with a standard deviation of $2\pi/15$ to the actions. The results, presented in Figure 8b, indicates an increase in the SAP metric, which is expected since a low SAP value is required for a well-disentangled LSBD representation. For the remaining metrics, GMA-VAE performs at least as well as the other methods, suggesting that it exhibits greater resilience to action noise.

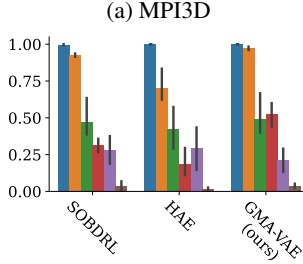

(a) MPI3D

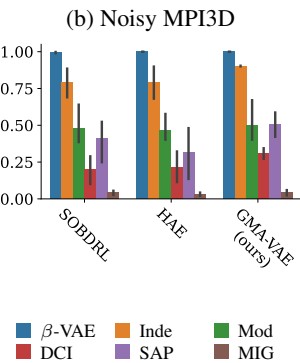

(b) Noisy MPI3D

Figure 8: Median of disentanglement metrics on MPI3D

## 6 ALTERNATIVE REPRESENTATION LEARNING PARADIGMS

This section shortly reviews three other related but different paradigms for representation learning. First, the causal representation learning approach (Schölkopf et al., 2021) proposes to ground the latent variables in the causal generative processes of the environment and seeks representations that correspond to underlying causal factors and their relations. Several identifiability results have been derived in this framework, relying on different assumptions, for instance regarding the available actions (or interventions) (Brehmer et al., 2022), the structure of the causal graph (Lippe et al., 2023), prior knowledge of the intervention targets (Lippe et al., 2022), or other inductive biases such as the sparsity of the causal graph (Lachapelle et al., 2022), or the transferability of causal representations (Bengio et al., 2020). Symmetry-based and causality-based disentanglement share some similarities (mathematically grounded, exploit interventions or actions) but have very different assumptions, justifying our choice not to include this framework in our comparisons.

Another line of work is object-centric representation learning, where the goal is to learn a factorized representation of objects and optionally their underlying dynamics. While promising, these methods often rely on either explicit structural assumptions or strong inductive biases that are closely tied to the nature of the observations, most notably images. In particular, they frequently exploits the spatial organization of visual scenes, encouraging representations that capture localized and compositional object structure (Zhu et al., 2018; Greff et al., 2019; Locatello et al., 2020; Kipf et al., 2022). Moreover, the question of disentangling the different features representing each object is often left aside. This field of research is not as focused on idenfiability proofs, but has shown strong empirical results in more complex and realistic environments.

A third line of work leverages group-equivariant neural networks, which exploit underlying symmetries. Classical approaches impose symmetries explicitly by designing architectures that are equivariant by construction (Cohen & Welling, 2016; Worrall et al., 2017; Weiler & Cesa, 2019; Dehmamy et al., 2021). However, these architectures are typically tailored to geometric transformations in the observation space (e.g., rotations or translations). More recent methods aim to learn equivariant neural networks with respect to more general symmetries, primarily by incorporating an additional equivariant loss term. In most cases, however, at least part of the group structure is assumed to be known (Mondal et al., 2022; Park et al., 2022; Shakerinava et al., 2022; Jin et al., 2024). While such symmetry-based approaches enhance the structure of the latent space and often improve sample efficiency and generalization, they do not explicitly focus on learning disentangled representations.

## 7 CONCLUSION

We introduced two independent algorithms: an action clustering method based on A-VAE, which provably recovers the ground-truth symmetry group structure, and a symmetry-based disentangled representation learning method, GMA-VAE, which achieves performance comparable to LSBD-VAE, even though the latter assumes prior knowledge of the action representations. Both of our methods rely on a strong assumption which requires the available actions to be disentangled. However, to the best of our knowledge, related state-of-the-art LSBD approaches also implicitly depend on this assumption to consistently learn a disentangled representation. While this restricts the applicability of the method to certain environments, it enables theoretical guarantees for both the action clustering and the disentanglement process. We further evaluate LSBD representations on downstream tasks and show that disentangled representations lead to significantly better long-term prediction performance and generalization, particularly in out-of-distribution scenarios.

A limitation of our approach compared to existing methods is that the full pipeline requires training two neural networks from scratch. A future work would be to initialize GMA-VAE with the pretrained encoder from A-VAE, or develop an end-to-end method that unifies the action clustering and representation learning steps into a single optimization process. Another limitation is the lack of more realistic experiments that explicitly challenge the assumptions of the LSBD framework and our own hypotheses. Investigating the behavior of our method under such conditions constitutes an important direction for future work.

## REPRODUCIBILITY STATEMENT

All the previous results are reproducible using the code on GitHub[1]. It includes all necessary components to generate the datasets, run the training procedures with the same hyperparameters and initialization seeds, and reproduce the figures.

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

## A  MATHEMATICAL BACKGROUND

**Vector Subspaces.**  Let $V$ be a vector space over a field $\mathbb{R}$. A subset $W \subseteq V$ is called a *vector subspace* if it is closed under vector addition and scalar multiplication; that is, for all $u, v \in W$ and $\lambda \in \mathbb{R}$, we have $u + v \in W$ and $\lambda u \in W$.

**Direct Sums of Subspaces.**  Let $V$ be a vector space and let $W_1, W_2 \subseteq V$ be subspaces. We say that $V$ is the *direct sum* of $W_1$ and $W_2$, denoted $V = W_1 \oplus W_2$, if every $v \in V$ can be uniquely written as $v = w_1 + w_2$ with $w_1 \in W_1$ and $w_2 \in W_2$, and $W_1 \cap W_2 = \{0\}$.

**Eigenvalues and Eigenspaces.**  Let $T : V \to V$ be a linear operator on a vector space $V$. A scalar $\lambda \in \mathbb{R}$ is an *eigenvalue* of $T$ if there exists a non-zero vector $v \in V$ such that $T(v) = \lambda v$. The corresponding set of vectors $\{v \in V \mid T(v) = \lambda v\}$ is called the *eigenspace* associated with $\lambda$, and is a subspace of $V$. The set of eignvalues is called the *spectrum*.

**Groups.**  A *group* is a set $G$ equipped with a binary operation $(x, y) \mapsto xy$ satisfying the following axioms:

- (Associativity) $(xy)z = x(yz)$ for all $x, y, z \in G$;
- (Identity element) There exists an element $e \in G$ such that $ex = xe = x$ for all $x \in G$;
- (Inverse element) For every $x \in G$, there exists $x^{-1} \in G$ such that $xx^{-1} = x^{-1}x = e$.

We often denote by $G^* = G \backslash \{e\}$ the set of non-identity elements of $G$.

**Examples of Groups.**

- The cyclic group $\mathbb{Z}/n\mathbb{Z}$ of integers modulo $n$.
- The symmetric group $\mathfrak{S}_n$ of permutations of $n$ elements.
- The general linear group $GL(\mathcal{V})$ of invertible linear transformations on a vector space $\mathcal{V}$.
- The special orthogonal group $SO(n)$ of $n \times n$ orthogonal matrices with determinant 1.

**Direct Product of Groups.**  Given two groups $G_1$ and $G_2$, their *direct product* is the group $G = G_1 \times G_2$ with the operation defined componentwise:

$$(g_1, g_2)(h_1, h_2) := (g_1 h_1, g_2 h_2).$$

Each group $G_i$ is referred to as a *direct factor* of $G$. By abuse of notation, $G_1$ is often identified with the subgroup $G_1 \times \{e_2\} \subseteq G$, where $e_2$ is the identity element of $G_2$.

**Subgroup and submonoid generated.** For a subset $S \subset G$, we call the *subgroup generated* by $S$ noted $\langle S \rangle$ the smallest subgroup of $G$ that contains $S$. We have $\langle S \rangle = \{s_1^{\varepsilon_1} s_2^{\varepsilon_2} \cdots s_k^{\varepsilon_k} \mid k \in \mathbb{N}, s_i \in S, \varepsilon_i \in \{\pm 1\}\}$. We call the *submonoid generated* by $S$ the set of all finite products of elements of $S$, and note it $\langle S \rangle_+ = \{s_1 s_2 \cdots s_k \mid k \in \mathbb{N}, s_i \in S\}$. If $G$ is finite, then for all $S \subset G$ we have $\langle S \rangle = \langle S \rangle_+$.

**Group Representations.** A *representation* of a group $G$ on a vector space $V$ is a map

$$\rho : G \times V \to V$$

such that for all $g, h \in G$ and $v \in V$, by denoting $g \cdot v$ for $\rho(g, v)$ we have

$$e \cdot v = v \quad \text{and} \quad g \cdot (h \cdot v) = (gh) \cdot v$$

Equivalently, a representation can be described as a group homomorphism

$$\rho : G \to GL(V), \quad \text{where } \rho(g)(v) := \rho(g, v).$$

The representation $\rho : G \to GL(V)$ is injective if and only if its kernel, defined as $\ker(\rho) := \{g \in G \mid \rho(g) = \mathrm{Id}_V\}$, is reduced to the identity element $\{e\}$. In this case, we write $\rho : G \hookrightarrow GL(V)$ and refer to it as a *faithful representation*.

**Direct Sum of Representations.** Let $(\rho_1, V_1)$ and $(\rho_2, V_2)$ be two representations of a group $G$. Their *direct sum* is the representation

$$\rho_1 \oplus \rho_2 : G \to GL(V_1 \oplus V_2)$$

defined by

$$(\rho_1 \oplus \rho_2)(g)(v_1, v_2) := (\rho_1(g)v_1, \rho_2(g)v_2).$$

The space $V_1 \oplus V_2$ is then said to carry the direct sum representation of $G$.

| Symbol | Meaning | Symbol | Meaning |
|---|---|---|---|
| $\mathcal{W}$ | World space | $\mathcal{Z}_1 \oplus \mathcal{Z}_2$ | Direct sum of vector spaces |
| $\mathcal{X}$ | Observation space | $\mathcal{A} \hookrightarrow \mathcal{B}$ | Injective map from $\mathcal{A}$ to $\mathcal{B}$ |
| $X$ | Observation random variable | $\|\|A\|\|$ | Spectral norm of matrix $A$ |
| $\mathcal{Z}$ | Latent space | $\mathbb{Z}/n\mathbb{Z}$ | Cyclic group of order $n$ |
| $Z$ | Latent random variable | $\mathfrak{S}_n$ | Symmetric group on $n$ elements |
| $G$ | Action Group | $GL(\mathcal{V})$ | General linear group on $\mathcal{V}$ |
| $\mathcal{G} \subset G$ | Available action set | $SO(\mathcal{V})$ | Special orthogonal group on $\mathcal{V}$ |
| $b : \mathcal{W} \to \mathcal{X}$ | Generative function | $\langle S \rangle$ | Group generated by set $S$ |
| $h : \mathcal{X} \to \mathcal{Z}$ | Encoder function | $\langle S \rangle_+$ | Monoid generated by set $S$ |
| $f = h \circ b : \mathcal{W} \to \mathcal{Z}$ | Latent generative function | $G^*$ | $G$ without its identity element |
| $\rho : G \to GL(\mathcal{Z})$ | Action representation | | |
| $\cdot_{\mathcal{Z}} : G \times \mathcal{Z} \to \mathcal{Z}$ | Group action on latent space | | |
| $\cdot_{\mathcal{W}} : G \times \mathcal{W} \to \mathcal{W}$ | Group action on world space | | |

Table 2: Table of symbols

# B   ELBO DERIVATION

We focus on the transition $\tau = (x, g, x')$. We have:

- $p_\theta(X'|z') = \mathcal{N}\big(\mu_\theta(z'), Diag(\sigma_\theta(z')^2)\big)$

- $p_{\psi,\phi}(Z'|x, g) = \mathcal{N}\big(\rho_\psi(g)\mu_\phi(x), Id\big)$

- $q_\phi(Z'|x') = \mathcal{N}\big(\mu_\phi(x'), Diag(\sigma_\phi(x')^2)\big)$

We will use several times the fact that if $Y \sim \mathcal{N}(\mu, Diag(\sigma)^2)$ then $\log p(y) = -\sum_i \log \sigma_i - \frac{1}{2}\left\|\frac{y-\mu}{\sigma}\right\|_2^2 + cste$ with the element-wise division in the norm.

Our initial goal is to optimize the log-likelihood $\log p(\tau) = \log p_{\psi,\phi}(x, g, x') = \log p_{\psi,\phi}(x'|x, g) + cst$. Indeed, we consider that the model has no prior (or a constant prior) on $(x, g)$ and therefore focus on optimizing the log-likelihood $\log p_{\psi,\phi}(x'|x, g)$:

$$
\begin{aligned}
\log p_{\psi,\phi}(x' \mid x, g) &= \mathbb{E}_{z' \sim q_\phi(z'|x')} \left[ \log \left( p_{\psi,\phi}(x' \mid x, g) \frac{q_\phi(z' \mid x')}{q_\phi(z' \mid x')} \right) \right] \\
&= \mathbb{E}_{z' \sim q_\phi(z'|x')} \left[ \log \left( \frac{p(x' \mid z', x, g) p_{\psi,\phi}(z' \mid x, g)}{p(z' \mid x', x, g)} \cdot \frac{q_\phi(z' \mid x')}{q_\phi(z' \mid x')} \right) \right] \\
&\qquad \text{According to Bayes formula} \\
&= \mathbb{E}_{z' \sim q_\phi(z'|x')} \left[ \log \frac{q_\phi(z' \mid x')}{p(z' \mid x', x, g)} \right] \\
&\quad + \mathbb{E}_{z' \sim q_\phi(z'|x')} \left[ \log \frac{p_{\psi,\phi}(z' \mid x, g)}{q_\phi(z' \mid x')} \right] \\
&\quad + \mathbb{E}_{z' \sim q_\phi(z'|x')} \left[ \log p(x' \mid z', x, g) \right] \\[2mm]
&= D_{KL} \left( q_\phi(z' \mid x') \| p(z' \mid x', x, g) \right) \quad \} \quad \geqslant 0 \\
&\quad - D_{KL} \left( q_\phi(z' \mid x') \| p_{\psi,\phi}(z' \mid x, g) \right) \\
&\quad + \mathbb{E}_{z' \sim q_\phi(z'|x')} \left[ \log p_\theta(x' \mid z') \right] \quad \text{According to Figure 3} \\[2mm]
&\geqslant -D_{KL} \left( q_\phi(z' \mid x') \| p_{\psi,\phi}(z' \mid x, g) \right) \\
&\quad + \mathbb{E}_{z' \sim q_\phi(z'|x')} \left[ \log p_\theta(x' \mid z') \right]
\end{aligned}
$$

The lower bound we have derived is composed of two lines. The first line corresponds to the KL divergence between two multivariate normal distributions and thus has an analytical expression. We have (up to an additive constant):

$$
-D_{KL} \left( q_\phi(z' \mid x') \| p_{\psi,\phi}(z' \mid x, g) \right) = -\frac{1}{2} \left\| \rho_\psi(g)\mu_\phi(x) - \mu_\phi(x') \right\|^2 - \frac{1}{2} \left\| \sigma_\phi(x') \right\|^2 + \sum_i \log \sigma_\phi(x')_i
$$

The second line is equal to (up to an additive constant):

$$
-\mathbb{E}_{z' \sim q_\phi(z'|x')} \left[ \sum_i \log \sigma_\theta(z')_i + \left\| \frac{x' - \mu_\theta(z')}{\sigma_\theta(z')} \right\|^2 \right]
$$

Putting everything together, we obtain:

$$
\begin{aligned}
\log p(\tau) \geqslant &-\frac{1}{2} \left\| \rho_\psi(g)\mu_\phi(x) - \mu_\phi(x') \right\|^2 - \frac{1}{2} \left\| \sigma_\phi(x') \right\|^2 + \sum_i \log \sigma_\phi(x')_i \quad \} \text{ action part} \\
&- \mathbb{E}_{z' \sim q_\phi(z'|x')} \left[ \sum_i \log \sigma_\theta(z')_i + \left\| \frac{x' - \mu_\theta(z')}{\sigma_\theta(z')} \right\|^2 \right] \qquad \} \text{ reconstruction part} \\
&+ C
\end{aligned}
$$

## C  ACTION CLUSTERING ALGORITHM AND PROOF OF THEOREM 2

Let $\mathcal{G} = \mathcal{G}_1 \cup \cdots \cup \mathcal{G}_K \subset G$ with $\forall k \; \mathcal{G}_k \subset G_k$ denote the ground-truth decomposition of the available action set. Our objective is to design an algorithm that recovers this decomposition based on $d_G$.

## C.1 ALGORITHM

Let $A \in \mathbb{R}^{d \times d}$, we denote by $\|A\| = \max_{z \in \mathbb{R}^d \setminus \{0\}} \|Az\| / \|z\|$ the spectral norm. We chose the following algorithm:

- Compute the variables
    - $r = \max_{g \in \mathcal{G}} \{\|A_g\|\}$ the maximal spectral norm,
    - $\varepsilon = \max_{(w,g,w') \in \mathcal{D}} \|A_g f(w) - f(w')\|$ the action loss upperbound,
    - $\eta = \varepsilon \left( 1 + \sum_{i=0}^{M} r^i \right)$ the threshold
- Start with unitary clusters: $\hat{K} = |\mathcal{G}|$ and $\hat{\mathcal{G}}_i = \{g_i\}$
- Iteratively merge the clusters $i$ and $j$ minimizing their distance

$$d(\hat{\mathcal{G}}_i, \hat{\mathcal{G}}_j) := \max_{g_i \in \hat{\mathcal{G}}_i; g_j \in \hat{\mathcal{G}}_j} d_G(g_i, g_j)$$

- Stop whenever the distance is above the threshold $\eta$.

What if the identity action $e$ belong the available action set $\mathcal{G}$ ? After a succesful convergence of the method, for all $g \in \mathcal{G}$ we have $d_G(e, g) \approx 0$ since $g = g^1 e$. As a result, $e$ is merged with another element at the first iteration and it will not influence the following computations of $d(\hat{\mathcal{G}}_i, \hat{\mathcal{G}}_j)$. As $e$ can be assigned to any subgroup, its presence does not impact on the performance or correctness of the overall method.

## C.2 PROOF OF CONVERGENCE

We aim to show that if Assumption 1 to 3 are satisfied, the dataset contains all the transitions, $\mathcal{W}$ is finite and A-VAE loss converge toward its minimum, then clustering algorithm will necessarily find the ground-truth decomposition at some point of the training.

For clarity, we assume that every composition of $d_G$ in Assumption 3 is of the form $g = u^m g'$. The proves can easily be adapted for the other forms.

**Proposition 1.** *Under Assumptions 2 and 3, two different actions $g, g' \in \mathcal{G}$ belong to the same subgroup if and only if there exists $m \in [\![1, M]\!]$ and $u \in \mathcal{G}$ such that $g = u^m g'$.*

*Proof.* The forward implication is given by Assumption 3. For the backward implication we distinguish two cases:

1. If one of element is the identity action $e$, then $e$ belong the same subgroup of every action

2. If both elements differ from $e$, then according to Assumption 2 there exists $a$, $b$ and $c$ such that $g \in G_a^*$, $g' \in G_b^*$ and $u^m \in G_c$. Therefore $g \in G_c \times G_b^* \setminus \{e\}$ and then $G_a \cap (G_c \times G_b) \neq \{e\}$, if we had $a \neq b$, this would contradict the direct decomposition of $G$.

$\square$

If the standard deviation of noises of A-VAE are fixed, then the loss for a transition $(x, g, x')$ is equal to

$$\mathcal{L} = \lambda_{ACT} \left\| \rho_\psi(g) \mu_\phi(x) - \mu_\phi(x') \right\|^2 + \mathbb{E}_{z' \sim q_\phi(x')} \left[ \left\| x' - \mu_\theta(z') \right\|^2 \right]$$

Unlike a $\beta$-VAE, which requires a trade-off between the regularization and the reconstruction, this loss can have both the action loss and the reconstruction loss converging toward 0.

When the action loss converges toward zero, we straightforwardly have $\varepsilon :=$ $\max_{(w,g) \in \mathcal{W} \times \mathcal{G}} \|A_g f(w) - f(g \cdot w)\| \to 0$ as it is its upper-bound. Additionally, since the coefficients of the action matrices are bounded in our implementation, $r$ is also bounded. As a result, the term $\eta$ converges toward zero during training.

We also have the following result:

**Proposition 2.** *Under Assumption 1, if the reconstruction loss converges toward zero, then*

$$\delta := \min_{w \neq w'} \|f(w) - f(w')\| \to +\infty$$

*Proof.* Let $h : \mathcal{X} \to \mathcal{Z}$ be the encoder and $d : \mathcal{Z} \to X$ be the decoder. Let $w_1 \neq w_2 \in \mathcal{W}$ with $x_i = b(w_i)$ and $z_i = h(x_i)$, as $b$ is injective we have $x_1 \neq x_2$. As the reconstruction loss converges toward $0$ and the dataset contains all the transitions and therefore all the observations, we have

$$\mathbb{E}_{z \sim \mathcal{N}(z_i, \sigma^2 I)} \left[ \|x_i - d(z)\|^2 \right] \to 0$$

Let denote $z^* = (z_1 + z_2)/2$, $\Delta = \|z_1 - z_2\|$, $B = B(z^*, 1)$ the ball of radius 1 centered at $z^*$ and $V(d)$ its volume. Let us denote $p(R) = \frac{1}{(\sqrt{2\pi}\sigma)^d} \exp\left(-\frac{R^2}{2\sigma^2}\right)$ the minimum value of the Gaussian density over a ball of radius $R$. We aim to show that if the reconstruction loss is sufficiently low, then $\Delta$ must be sufficiently large for the contribution of the reconstruction over the ball to become negligible.

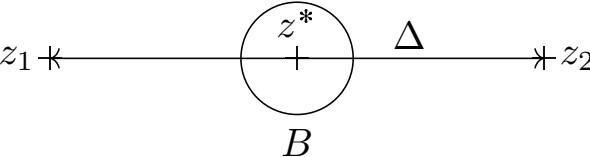

Since convergence in $L^2$ in probability implies convergence in $L^1$ in probability, for any $\epsilon > 0$, at some point of the training there is

$$\int_{z \in \mathbb{R}^d} \mathcal{N}(z; z_i, \sigma^2 I) \|x_i - d(z)\| dz \leqslant \epsilon$$

We have

$$\epsilon \geqslant \int_{z \in \mathbb{R}^d} \mathcal{N}(z; z_i, \sigma^2 I) \|x_i - d(z)\| dz$$

$$\geqslant \int_{z \in B} \mathcal{N}(z; z_i, \sigma^2 I) \|x_i - d(z)\| dz$$

$$\geqslant p\left(\frac{\Delta}{2} + 1\right) \int_{z \in B} \|x_i - d(z)\| dz$$

by definition of $p$, since for any $z \in B$, $\|z_i - z\| \leqslant \|z_i - z^*\| + \|z^* - z\| \leqslant \Delta/2 + 1$

Therefore, by summing the two equations, we obtain:

$$2\epsilon \geqslant p\left(\frac{\Delta}{2} + 1\right) \int_{z \in B} (\|x_1 - d(z)\| + \|x_2 - d(z)\|) \, dz$$

$$\geqslant p\left(\frac{\Delta}{2} + 1\right) \int_{z \in B} \|x_1 - x_2\| dz \qquad \text{with triangular inequality}$$

$$\geqslant p\left(\frac{\Delta}{2} + 1\right) V(d) \|x_1 - x_2\|$$

Using the definition of $p$ and isolating $\Delta = \|z_1 - z_2\|$, we obtain

$$\|z_1 - z_2\| \geqslant 2\sigma \sqrt{2\log\left(\frac{1}{\epsilon}\frac{V(d)\|x_1 - x_2\|}{2(\sqrt{2\pi}\sigma)^d}\right) - 2}$$
$$\xrightarrow[\epsilon \to 0]{} +\infty$$

Therefore $\|z_1 - z_2\| \to +\infty$ as the reconstruction loss converges toward 0. Finally, as $\mathcal{W}$ is finite, we have $\min_{w \neq w'} \|f(w) - f(w')\| \to +\infty$.

$\square$

Consequently, at some point of the training, the inequality $\delta > 2\eta$ holds.

**Proposition 3.** *Under Assumptions 2 and 3, if $\delta > 2\eta$ then $g$ and $g'$ belong to the same subgroup if and only if $d_G(g, g') \leqslant \eta$*

*Proof.* Suppose that $g, g' \in \mathcal{G}$ with $g \neq g'$ belong to the same subgroup, therefore, according to Proposition 1, there exists $u \in \mathcal{G}$ and $m \in [\![1, M]\!]$ such that $g = u^m g'$. Let $w \in W$

$$\begin{aligned}
\|A_g f(w) - A_u^m A_{g'} f(w)\| &\leqslant \|A_g f(w) - f(g \cdot w)\| \\
&\quad + \|f(u^m g' \cdot w) - A_u f(u^{m-1} g' \cdot w)\| \\
&\quad + \cdots \\
&\quad + \|A_u^{m-1} f(u g' \cdot w) - A_u^m f(g' \cdot w)\| \\
&\quad + \|A_u^m f(g' \cdot w) - A_u^m A_{g'} f(w)\| \\
&\leqslant \|A_g f(w) - f(g \cdot w)\| \\
&\quad + \|f(u^m g' \cdot w) - A_u f(u^{m-1} g' \cdot w)\| \\
&\quad + \cdots \\
&\quad + \|A_u\|^{m-1} \cdot \|f(u g' \cdot w) - A_u f(g' \cdot w)\| \\
&\quad + \|A_u\|^m \cdot \|f(g' \cdot w) - A_{g'} f(w)\| \\
&\leqslant \varepsilon + \sum_{i=0}^{m} r^i \varepsilon \\
&\leqslant \eta
\end{aligned}$$

After applying expectation over $w$ we find $d_G(g, g') \leqslant \|A_g - A_u^m A_{g'}\|_h \leqslant \eta$

Let us now suppose that $g, g' \in \mathcal{G}$ do not belong to same subgroup, therefore, according to Proposition 1, for all $u \in \mathcal{G}$ and $m \in [\![1, M]\!]$ we have

$$\begin{aligned}
\delta &\leqslant \|f(g \cdot w) - f(u^m g' \cdot w)\| \\
&\leqslant \|f(g \cdot w) - A_g f(w)\| \\
&\quad + \|A_g f(w) - A_u^m A_{g'} f(w)\| \\
&\quad + \|A_u^m A_{g'} f(w) - f(u^m g' \cdot w)\| \\
&\leqslant \varepsilon + \|A_g f(w) - A_u^m A_{g'} f(w)\| + \sum_{i=0}^{m} r^i \varepsilon \\
&\leqslant \|A_g f(w) - A_u^m A_{g'} f(w)\| + \eta
\end{aligned}$$

Therefore $\|A_g f(w) - A_u^m A_{g'} f(w)\| \geqslant \delta - \eta > \eta$, after applying expectation over $w$ and $\min$ over $u$ and $m$ we get $d_G(g, g') > \eta$

$\square$

Therefore, at some point of the training, two actions belong to the same subgroup if and only if their distance with repect to $d_G$ is below the threshold $\eta$. As a consequence, the clustering algorithm described previously successfully recovers the ground-truth decomposition, hence proving Theorem 2.

In practice we found that using $\eta = \sigma$ the fixed latent noise standard deviation as a threshold yielded better empirical results. Consequently, we use this value for $\eta$ in all our experiments.

## D  PROOF OF THEOREM 3

If the standard deviation of noises of GMA-VAE are fixed, then the loss for a transition $(x, g, x')$ equals:

$$\mathcal{L}(x, g, x') = \lambda_{DIS} \sum_i \mathcal{H}(\pi_{:,i}) + \lambda_{ACT} \left\| \rho_\psi(g) \mu_\phi(x) - \mu_\phi(x') \right\|^2 + \mathbb{E}_{z' \sim q_\phi(x')} \left[ \left\| x' - \mu_\theta(z') \right\|^2 \right]$$

Let us suppose that $G$ is finite, that the dataset contains all possible transitions, and that the losses of GMA-VAE have converged to their global minimum of 0. We aim to prove that the encoder is a LSBD representation.

We cannot directly prove that it is disentangled with respect to $\langle \mathcal{W}, b, \prod_k G_k \rangle$, as we cannot build a representation of an action that is not generated by the available action set. Instead, we aim to prove that the encoder is disentangled with respect to $\langle \mathcal{W}, b, \prod_k \langle \mathcal{G}_k \rangle \rangle$ with $\langle \mathcal{G}_k \rangle$ being the subgroup generated by $\mathcal{G}_k$. Similarly, we cannot prove that it is disentangled over all $\mathcal{Z}$. Therefore, as done by Keurti et al. (2023), we restrict the latent space to $\mathcal{V} = \text{span}(f(\mathcal{W})) \subset \mathcal{Z}$

We proceed by proving that the learned representation satisfies all the criteria listed in Definition 1:

*(1) There exists a group action $\cdot_{\mathcal{Z}} : G \times \mathcal{V} \to \mathcal{V}$*

First, note that $\prod_k \langle \mathcal{G}_k \rangle = \langle \mathcal{G} \rangle$. Let $g \in \langle \mathcal{G} \rangle$, as $G$ is finite, we can write $g = g^{(1)} \cdots g^{(n)}$ with $\forall i,\ g^{(i)} \in \mathcal{G}$, the group action is given by:

$$g \cdot_{\mathcal{Z}} z = \rho(g)z \text{ with } \rho(g) = \prod_i \widetilde{A}_{g^{(i)}}$$

Note that the definition of $\rho(g)$ depends on the decomposition of $g$, which is not necessarily unique. Since any decomposition would be satisfying and $\mathcal{G}$ is finite, we can arbitrarily chose one decomposition for each $g$.

The fact that $\cdot_{\mathcal{Z}}$ is a group action is given thanks to the equivariance and is proven below.

*(2) Equivariance holds:* $\forall (g, w) \in \langle \mathcal{G} \rangle \times \mathcal{W},\ g \cdot_{\mathcal{Z}} f(w) = f(g \cdot_{\mathcal{W}} w)$

As the action loss is equal to zero we have:

$$\forall (g, w) \in \mathcal{G} \times \mathcal{W},\ g \cdot_{\mathcal{Z}} f(w) = f(g \cdot_{\mathcal{W}} w)$$

Let $g = \prod_i g^{(i)} \in \langle \mathcal{G} \rangle$ with $g^{(i)} \in \mathcal{G}$ and $w \in \mathcal{W}$. We apply recursively the previous equivariance to get $g \cdot_{\mathcal{Z}} f(w) = f(g \cdot_{\mathcal{W}} w)$, hence proving the equivariance over all $\langle \mathcal{G} \rangle \times \mathcal{W}$.

We now prove that $\cdot_{\mathcal{Z}}$ is indeed a group action thanks to the equivariance property. For all $w \in \mathcal{W}$ and $g, g' \in \langle \mathcal{G} \rangle$:

- $g \cdot_{\mathcal{Z}} f(w) = f(g \cdot_{\mathcal{W}} w) \in \mathcal{V}$
- $e \cdot_{\mathcal{Z}} f(w) = f(e \cdot_{\mathcal{W}} w) = f(w)$
- $g' \cdot_{\mathcal{Z}} (g \cdot_{\mathcal{Z}} f(w)) = g' \cdot_{\mathcal{Z}} f(g \cdot_{\mathcal{W}} w)$
$$= f(g' \cdot_{\mathcal{W}} (g \cdot_{\mathcal{W}} w))$$
$$= f((g'g) \cdot_{\mathcal{W}} w)$$
$$= (g'g) \cdot_{\mathcal{Z}} f(w)$$

As $f(\mathcal{W})$ generates $\mathcal{V}$, for all $z \in \mathcal{V}$ and $g, g' \in \langle \mathcal{G} \rangle$ we have:

- $g \cdot_{\mathcal{Z}} z \in \mathcal{V}$
- $e \cdot_{\mathcal{Z}} z = z$
- $g' \cdot_{\mathcal{Z}} (g \cdot_{\mathcal{Z}} z) = (g'g) \cdot_{\mathcal{Z}} z$

*(3) There exists a decomposition $\mathcal{V} = \mathcal{V}_1 \oplus \cdots \oplus \mathcal{V}_K$ and group actions $\cdot_k : G_k \times \mathcal{V}_k \to \mathcal{V}_k$ such that:*

$$(g_1, \ldots, g_K) \cdot_{\mathcal{Z}} (z_1, \ldots, z_K) = (g_1 \cdot_1 z_1, \ldots, g_K \cdot_K z_K)$$

As the disentanglement loss is equal to zero, the masks are binary *i.e.* $\pi_k \in \{0, 1\}^d$. Therefore the matrices $\tilde{A}_g$ for $g \in \mathcal{G}$ satisfy the block structure illustrated in Figure 5. Additionally if $g_k \in \langle \mathcal{G}_k \rangle$ it can be decomposed into $g_k = g_k^{(1)} \cdots g_k^{(n_k)}$ with $g_k^{(i)} \in \mathcal{G}_k$ and therefore $\rho(g_k) = \prod_i \tilde{A}_{g_k^{(i)}}$ share the same block structure. Finally, for each $g = (g_1, \ldots, g_K) \in \langle \mathcal{G} \rangle$, $\rho(g) = \prod_k \rho(g_k)$ is block diagonal up to a permutation of the indices.

Let us first find a decomposition and group actions over $\mathcal{Z}$. We take $\mathcal{Z}_k = \text{span}\{e_i \mid \pi_{k,i} = 1\}$ with $e_i$ the standard basis vectors, this choice reflects the objective of achieving disentanglement along the Cartesian axes. Hence we have $\mathcal{Z} = \mathcal{Z}_1 \oplus \cdots \oplus \mathcal{Z}_K$ and $z_k = \pi_k \odot z$.

Additionally we take $g_k \cdot_k z_k = \rho(g_k) z_k$, we have $\rho(g_k) z_k \in \mathcal{Z}_k$ as $\rho(g_k)$ is the identity on the complement subspace of $\mathcal{Z}_k$. We therefore have for each $g = (g_1, \ldots, g_K) \in \langle \mathcal{G} \rangle$ and $z \in \mathcal{Z}$:

$$g \cdot_{\mathcal{Z}} z = \rho(g) z$$

$$= \rho(g) \left( \sum_k z_k \right)$$

$$= \sum_k \rho(g) z_k$$

$$= \sum_k \rho(g_k) z_k$$

$$= \sum_k g_k \cdot_k z_k$$

$$= (g_1 \cdot_1 z_1, \ldots, g_K \cdot_K z_K)$$

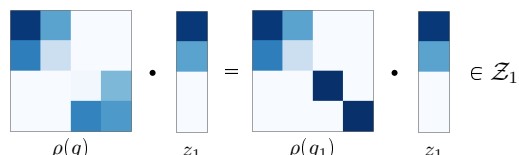

Figure 9: Matrix representation of $\rho(g) z_k = \rho(g_k) z_k \in \mathcal{Z}_k$, here the two first dimensions corresponds to $\mathcal{Z}_1$ and the remaining two to $\mathcal{Z}_2$.

Let us prove that the decomposition and group actions restricted on $\mathcal{V}$ are satisfying. Let's take $\mathcal{V}_k = \text{span}\{f(w)_k \mid w \in \mathcal{W}\} = \mathcal{V} \cap \mathcal{Z}_k$ with $f(w)_k$ the projection of $f(w)$ on $\mathcal{Z}_k$, therefore the $\mathcal{V}_k$ are in direct sum. Moreover we have for all $w \in \mathcal{W}$, $f(w) = \sum_k f(w)_k$ with $f(w)_k \in \mathcal{V}_k$ and therefore $\mathcal{V} = \mathcal{V}_1 \oplus \cdots \oplus \mathcal{V}_K$.

Additionally, for all $g \in \langle \mathcal{G} \rangle$ and $f(w) \in \mathcal{V}$ we have $g \cdot_{\mathcal{Z}} f(w) = f(g \cdot_{\mathcal{W}} w)$ meaning that for all component $k$ we have $g_k \cdot_k z_k = f(g \cdot_{\mathcal{W}} w)_k \in \mathcal{V}_k$. Finally, as previously done for $\cdot_{\mathcal{Z}}$, it can be shown that $\cdot_k : G_k \times \mathcal{V}_k \to \mathcal{V}_k$ are group actions.

*(4) The representation $h$ is injective:*

As the reconstruction loss is equal to zero and Assumption 1 holds, the distance between the encoding of two world states has a lower bound as shown in the proof of Proposition 2.

*(5) The group action $\cdot_{\mathcal{Z}}$ is linear:*

As highlighted by Keurti et al. (2023), the restriction of $\rho$ on $\mathcal{V}$ written $\rho_{\mathcal{V}} : G \to GL(\mathcal{V})$ is a morphism and we have $g \cdot_{\mathcal{Z}} z = \rho_{\mathcal{V}}(g) z$. This can be proven similarly to the argument used to show that $\cdot_{\mathcal{Z}}$ is a group action. Hence proving the representation is linear on $\mathcal{V}$.

## E  MINIMAL LINEAR REPRESENTATION DIMENSION

Let $G$ a group, we seek the minimal dimension such that there exists an injective morphism into invertible matrices $\delta(G) = \min\{d \mid \exists \rho : G \hookrightarrow GL(\mathbb{R}^d)\}$. We also seek $\delta_{SO}(G)$ the minimal dimension for special orthogonal matrices $\delta_{SO}(G) = \min\{d \mid \exists \rho : G \hookrightarrow SO(d)\}$.

We get the following results:

| $G$ | $\delta$ | $\delta_{SO}$ |
|---|---|---|
| $\mathbb{Z}$ | 1 | 2 |
| $\mathbb{Z}/2\mathbb{Z}$ | 1 | 2 |
| $\mathbb{Z}/n\mathbb{Z}, n \geqslant 3$ | 2 | 2 |
| $\mathfrak{S}_2$ | 1 | 2 |
| $\mathfrak{S}_3$ | 2 | 3 |
| $\mathfrak{S}_4$ | 3 | 3 |
| $\mathfrak{S}_n, n \geqslant 5$ | $n-1$ | $n-1$ or $n$ |

## E.1 FINITE CYCLIC GROUP IE ROTATION GROUP

*Proof.* Let $G$ a finite cyclic group of cardinal $n \geqslant 2$ *i.e.* $G \cong \mathbb{Z}/n\mathbb{Z}$. There are two cases to consider:

$\underline{n = 2}$: Then $G = \{e, g\}$ with $g^2 = e$. Therefore $\delta(G) = 1$ because the isomorphism $\rho = \{e \mapsto (1); \ g \mapsto (-1)\}$ is satisfying. Furthermore $\delta_{SO}(G) > 1$ as $SO(1) = \{(1)\}$ can only express the trivial group.

$\underline{n > 2}$: Suppose $\delta(G) = 1$, then for each element $g \in G$ there exists a scalar $\lambda_g \in \mathbb{R}$ such that $\rho(g) = (\lambda_g)$. Therefore for all $g$ and $k \geqslant 0$ we have $g^k \in G$ and then $\lambda_g^k \in \{\lambda_{g'} \mid g' \in G\}$ which is finite set. Therefore $\lambda_g \in \{-1, 1\}$, consequently for all $g \in G$, $\rho(g^2) = (\lambda_g^2) = (1)$ and then $g^2 = e$. This is contradictory, therefore $\delta(G) \geqslant 2$.

We show that $\delta(G) = 2$: let $h$ be a generator of $G$ *i.e.* for all $g \in G$, there exists $k_g \in [\![0, n[\![$ such that $h^{k_g} = g$. The following rotation matrix is a satisfying morphism $\rho(g) = \begin{pmatrix} \cos(2\pi k_g/n) & -\sin(2\pi k_g/n) \\ \sin(2\pi k_g/n) & \cos(2\pi k_g/n) \end{pmatrix}$. Finally $\delta(G) = 2$ and therefore $\delta_{SO}(G) = 2$ □

## E.2 INFINITE CYCLIC GROUP

*Proof.* Let $G$ be an infinite cyclic group *i.e.* $G \cong \mathbb{Z}$. Let $h$ be a generator $G$, therefore for all $g \in G$, there exists $k_g \in \mathbb{Z}$ such that $g = h^{k_g}$. For any $x > 0$, $x \neq 1$, the representation $\rho : g \in G \mapsto (x^{k_g})$ is satisfying. Therefore $\delta(G) = 1$.

As previously, we have $\delta_{SO}(G) > 1$ and for any $\theta \notin 2\pi\mathbb{Q}$, the representation $\rho(g) = \begin{pmatrix} \cos(k_g\theta) & -\sin(k_g\theta) \\ \sin(k_g\theta) & \cos(k_g\theta) \end{pmatrix} \in SO(2)$ is satisfying. Finally $\delta_{SO}(G) = 2$. □

## E.3 PERMUTATION GROUP

*Proof.* Let $G = \mathfrak{S}_n$ with $n \geqslant 2$ the permutation group.

Generalities: We know that there exists an injective morphism $\rho : \mathfrak{S}_n \hookrightarrow O(n-1)$ thanks to its standard representation, therefore $\delta(\mathfrak{S}_n) \leqslant n-1$. Furthermore we can inject $O(n-1)$ into $SO(n)$ using $g \mapsto \begin{pmatrix} \rho(g) & 0 \\ 0 & \det\rho(g) \end{pmatrix}$. And then $\delta_{SO}(\mathfrak{S}_n) \leqslant n$.

$\underline{n = 2}$: We have $\mathfrak{S}_2 \cong \mathbb{Z}/2\mathbb{Z}$, as previously $\delta(\mathfrak{S}_2) = 1$ and $\delta_{SO}(\mathfrak{S}_2) = 2$.

$\underline{n = 3}$: Similarly to previously $\delta(\mathfrak{S}_3) > 1$ and $\delta_{SO}(\mathfrak{S}_3) > 1$. Moreover there is no injection $\mathfrak{S}_3 \hookrightarrow SO(2)$ as $\mathfrak{S}_3$ would be commutative. Therefore $\delta(\mathfrak{S}_3) = 2$ and $\delta_{SO}(\mathfrak{S}_3) = 3$

$\underline{n = 4}$: $\mathfrak{S}_4$ is isomorphic to a subgroup of $SO(3)$ as it can be seen as the symmetry group of the tetrahedron. Moreover their is no injection $\mathfrak{S}_4 \hookrightarrow GL(\mathbb{R}^2)$ as $\mathfrak{S}_4$ would be cyclic or dihedral. Then $\delta(\mathfrak{S}_4) = \delta_{SO}(\mathfrak{S}_4) = 3$.

$\underline{n \geqslant 5}$: The minimal dimension for a linear representation in $\mathbb{C}$ of $\mathfrak{S}_n$ is $n-1$ (Rasala, 1977). Therefore the minimal dimension in $\mathbb{R}$ is at least $n-1$ *i.e.* $\delta(G) \geqslant n-1$. Consequently $\delta(\mathfrak{S}_n) = n-1$ and $\delta_{SO}(\mathfrak{S}_n) \in \{n-1, n\}$ □

## F    BLOCK-DIAGONAL PARAMETERIZATION IS NOT ENOUGH FOR LINEAR DISENTANGLEMENT

Here, we aim to show that using a structured action parameterization, as done in SOBDRL (Quessard et al., 2020), LSBD-VAE* and HAE (Keurti et al., 2023) is not sufficient to obtain a disentangled representation, even when applying the regularization used in SOBDRL. Assume that the underlying group action is composed of two cyclic direct factors, i.e., $G = G_1 \times G_2$, and that $G$ is isomorphic to a subgroup of $SO(2) \times SO(2)$. Our goal is to learn a disentangled representation of this subgroup. Moreover, we require the learned representation to be injective as any constant morphism would be disentangled. A block-diagonal-based method may use this prior knowledge by parameterizing the action matrices with $\rho : g \in \mathcal{G} \mapsto R(\theta_g, \theta'_g) := \begin{pmatrix} R_2(\theta_g) & 0 \\ 0 & R_2(\theta'_g) \end{pmatrix}$ with $R_2(\theta)$ being the rotation matrix of angle $\theta$.

Suppose that the unknown ground-truth group decomposition consists of two cyclic direct factors with $n$ and $m$ elements, i.e., $G = \mathbb{Z}/n\mathbb{Z} \times \mathbb{Z}/m\mathbb{Z}$. Let $g_1$ and $g_2$ denote generators of each respective factor. For the action representation to be disentangled, we would ideally want something like $\rho(g_1) = (2\pi/n, 0)$ and $\rho(g_2) = (0, 2\pi/m)$, so that each generator only affects a single latent subspace. However, when using this type of parameterization as a disentanglement criterion, even with the regularization term introduced in SOBDRL, it remains impossible to guarantee disentanglement of the action representation. Below, we present examples of entangled representations that satisfy the imposed parameterization but fail to be disentangled.

Case n°1: The representation such that $\rho(g_1) = R(2\pi/n, 0)$ and $\rho(g_2) = R(2\pi/m, 2\pi/m)$ is an injective homomorphism, it was encountered during our LSBD-VAE* experiments. This representation is entangled with respect to the LSBD framework.

*Proof.* Suppose there exists a decomposition $\mathbb{R}^4 = \mathcal{Z}_1 \oplus \mathcal{Z}_2$ satisfying the disentangled definition, then $\rho(g_2)$ would be the identity function on $Z_1$ *i.e.* for all $z_1 \in \mathcal{Z}_1$, $\rho(g_2)z_1 = z_1$. If $\mathcal{Z}_1 \neq \{0\}$ then 1 is an eigenvalue of $\rho(g_2)$ which is impossible as its spectrum is $\{e^{2i\pi/m}, e^{-2i\pi/m}\}$. Therefore $\mathcal{Z}_1 = \{0\}$, the same reasoning can be applied on $\mathcal{Z}_2$, therefore $\mathcal{Z}_1 \oplus \mathcal{Z}_2 = \{0\} \neq \mathbb{R}^4$. □

Case n°2 If $n$ and $m$ are coprime numbers, then the representation such that $\rho(g_1) = R(2\pi/n, 0)$ and $\rho(g_2) = R(2\pi/m, 0)$ is injective as

$$G = \frac{\mathbb{Z}}{n\mathbb{Z}} \times \frac{\mathbb{Z}}{m\mathbb{Z}} \cong \frac{\mathbb{Z}}{nm\mathbb{Z}}$$

This type of representation was encountered with SOBDRL as it minimizes its disentanglement criterion. However this representation is entangled with respect to the LSBD framework

*Proof.* Suppose there exists a decomposition $\mathbb{R}^4 = \mathcal{Z}_1 \oplus \mathcal{Z}_2$ respecting the definition, then $\rho(g_1)$ would be the identity on $\mathcal{Z}_2$ *i.e.* for all $z_2 \in \mathcal{Z}_2$, $\rho(g_1)z_2 = z_2$ and then $\mathcal{Z}_2$ is a subspace of the eigenspace of $\rho(g_1)$ associated with the eigenvalue 1. This eigenspace corresponds to the two last dimension: $\mathbb{R}^4$ *i.e.* $\mathcal{Z}_2 \subset \{0\} \times \{0\} \times \mathbb{R}^2$. Similarly for $\rho(g_2)$, we have $\mathcal{Z}_1 \subset \{0\} \times \{0\} \times \mathbb{R}^2$ and therefore $\mathbb{R}^4 \neq \mathcal{Z}_1 \oplus \mathcal{Z}_2$. □

Case n°3: If $n = m = pq$ a composite number with $p$ and $q$ coprimes, the representation $\rho(g_1) = R(2\pi/p, 2\pi/q)$ and $\rho(g_2) = R(2\pi/q, 2\pi/p)$ is injective. It is like switching the two $\mathbb{Z}/q\mathbb{Z}$ components of each direct factor.

$$G \cong \overbrace{\left( \frac{\mathbb{Z}}{p\mathbb{Z}} \times \frac{\mathbb{Z}}{q\mathbb{Z}} \right)}^{G_1} \times \overbrace{\left( \frac{\mathbb{Z}}{p\mathbb{Z}} \times \frac{\mathbb{Z}}{q\mathbb{Z}} \right)}^{G_2}$$

As for case n°1, this representation is entangled with respect to the LSBD framework

## G    PROOF OF THEOREM 1

**Reminders**:

- $\mathcal{W}$ the world state set
- $\mathcal{X}$ the observation set
- $\mathcal{Z} = \mathbb{R}^d$ latent space
- $b : \mathcal{W} \to \mathcal{X}$ the observation function
- $h : \mathcal{X} \to \mathcal{Z}$ the encoding function
- $f = h \circ b : \mathcal{W} \to \mathcal{Z}$
- $h$ is disentangled with respect to $\langle \mathcal{W}, b, \prod_k G_k \rangle$ if:
    1. There exists $\cdot_{\mathcal{Z}} : G \times \mathcal{Z} \to \mathcal{Z}$
    2. Equivariance holds: $\forall g \in G, w \in \mathcal{W}, g \cdot_{\mathcal{Z}} f(w) = f(g \cdot_{\mathcal{W}} w)$
    3. There exists $\mathcal{Z} = \mathcal{Z}_1 \oplus \cdots \oplus \mathcal{Z}_K$ and $\cdot_k : G_k \times \mathcal{Z}_k \to \mathcal{Z}_k$ such that
    $$(g_1, \ldots, g_K)(z_1, \ldots, z_K) = (g_1 \cdot_1 z_1, \ldots, g_K \cdot_K z_K)$$
    4. $h$ is injective

Suppose there exists a SBD representation with respect to $\langle \mathcal{W}, b, \prod_k G_k \rangle$, we aim to prove that we can build $\langle \widetilde{\mathcal{W}}, \tilde{b}, \prod_k G_k \rangle$ from $\langle W, b, \prod_k G_k \rangle$ such that (1) the SBD representations with respect to $\langle \widetilde{\mathcal{W}}, \tilde{b}, \prod_k G_k \rangle$ are exactly the SBD representations with respect to $\langle \mathcal{W}, b, \prod_k G_k \rangle$, (2) the observation function $\tilde{b}$ is injective and (3) $\langle W, b, \prod_k G_k \rangle$ and $\langle \widetilde{W}, \tilde{b}, \prod_k G_k \rangle$ yield the same sensorimotor interaction with the agent, and are thus indistinguishable.

### G.1    INTERACTION EQUIVALENCE

We would like to reduce $\mathcal{W}$ to indistinguishable cases, we therefore define the following equivalence relation:

**Definition 2.** *Two world states $w_1$ and $w_2$ are called interaction equivalent if and only if:*

$$\forall g \in G, b(g \cdot_{\mathcal{W}} w_1) = b(g \cdot_{\mathcal{W}} w_2)$$

*We denote this relation between the two world states as $w_1 \sim w_2$.*

This definition implies that two states $w_1$ and $w_2$ that yield the same observation but can be distinguished by interacting with the environment are not equivalent. Therefore, it does not cover situations such as object occlusions. Based on this interaction equivalence relation, we define a new set composed of the equivalence classes:

**Definition 3.** *We call interaction equivalent world state set and denote by $\widetilde{\mathcal{W}}$ the set of interaction equivalence classes.*

$$\widetilde{\mathcal{W}} = \{[w] \mid w \in \mathcal{W}\}$$

*where $[w]$ denotes the equivalence class of $w$ according to Definition 2.*

We now turn to defining appropriate functions using the interaction equivalent world state set.

**Proposition 4.** *If $w_1 \sim w_2$, then $b(w_1) = b(w_2)$.*

*Proof.* This can be easily derived from Definition 2 and taking $g = e$ the identity element.    $\square$

This proposition allows us to define a new observation function on the interaction equivalent world state set $\widetilde{\mathcal{W}}$:

**Definition 4.** *We call interaction equivalent observation function and denote by $\tilde{b}$ the function $\tilde{b} : \widetilde{\mathcal{W}} \to \mathcal{X}$ such that:*

$$\forall [w] \in \widetilde{\mathcal{W}}, \tilde{b}([w]) = b(w)$$

**Proposition 5.** *If $w_1 \sim w_2$, then $\forall g \in G, g \cdot_\mathcal{W} w_1 \sim g \cdot_\mathcal{W} w_2$.*

*Proof.* Let $w_1$ and $w_2$ be two world states from $\mathcal{W}$ such that $w_1 \sim w_2$. Let $g \in G$ be a symmetry. We aim to prove that $g \cdot_\mathcal{W} w_1 \sim g \cdot_\mathcal{W} w_2$.

Let $g' \in G$ be a symmetry. Since $G$ is a group, it follows that the composition $g'g \in G$. According to Definition 2:

$$b(g'g \cdot_\mathcal{W} w_1) = b(g'g \cdot_\mathcal{W} w_2)$$

And, by definition of a group action:

$$b(g' \cdot_\mathcal{W} (g \cdot_\mathcal{W} w_1)) = b(g' \cdot_\mathcal{W} (g \cdot_\mathcal{W} w_2))$$

According to Definition 2, we thus have $g \cdot_\mathcal{W} w_1 \sim g \cdot_\mathcal{W} w_2$. $\square$

This proposition allows us to define a new group action on the interaction equivalent world state set $\widetilde{\mathcal{W}}$:

**Definition 5.** *We call interaction equivalent group action and denote by $\cdot_{\widetilde{\mathcal{W}}}$ the function $\cdot_{\widetilde{\mathcal{W}}} : G \times \widetilde{\mathcal{W}} \to \widetilde{\mathcal{W}}$ such that:*

$$\forall g \in G, \forall [w] \in \widetilde{\mathcal{W}}, g \cdot_{\widetilde{\mathcal{W}}} [w] = [g \cdot_\mathcal{W} w]$$

**Proposition 6.** *$\cdot_{\widetilde{\mathcal{W}}}$ is a group action.*

*Proof.* We need to prove two properties:

1. $\forall [w] \in \widetilde{\mathcal{W}}, e \cdot_{\widetilde{\mathcal{W}}} [w] = [w]$

2. $\forall (g, g') \in G, \forall [w] \in \widetilde{\mathcal{W}}, g' \cdot_{\widetilde{\mathcal{W}}} (g \cdot_{\widetilde{\mathcal{W}}} [w]) = (g'g) \cdot_{\widetilde{\mathcal{W}}} [w]$

We start with the first property. Let $[w] \in \widetilde{\mathcal{W}}$ be an interaction equivalence class. We have:

$$\begin{aligned} e \cdot_{\widetilde{\mathcal{W}}} [w] &= [e \cdot_\mathcal{W} w] && \text{according to Definition 5} \\ &= [w] && \text{since } \cdot_\mathcal{W} \text{ is a group action itself} \end{aligned}$$

For the second property, suppose $(g, g') \in G$ and $[w] \in \widetilde{\mathcal{W}}$. We have:

$$\begin{aligned} g' \cdot_{\widetilde{\mathcal{W}}} (g \cdot_{\widetilde{\mathcal{W}}} [w]) &= g' \cdot_{\widetilde{\mathcal{W}}} [g \cdot_\mathcal{W} w] && \text{according to Definition 5} \\ &= [g' \cdot_\mathcal{W} (g \cdot_\mathcal{W} w)] && \text{according to Definition 5} \\ &= [(g'g) \cdot_\mathcal{W}] && \text{since } \cdot_\mathcal{W} \text{ is a group action} \\ &= (g'g) \cdot_{\widetilde{\mathcal{W}}} [w] && \text{according to Definition 5} \end{aligned}$$

$\square$

In summary, we have proposed an equivalence relation for world states that allows us to define a new world state set regrouping equivalent states, as well as an observation function and a group action for this new world state set.

## G.2 NECESSITY OF INJECTIVITY OF THE OBSERVATION FUNCTION

We now address the main point of this appendix, which is to show that if $h$ is an SBD representation with respect to $\langle \mathcal{W}, b, \prod_i G_i \rangle$, then $\tilde{b}$ has to be injective. We start by first showing that SBD-ness is implied in the interaction equivalent world.

**Proposition 7.** *If $h$ is SBD with respect to $\langle \mathcal{W}, b, \prod_i G_i \rangle$, then $h$ is SBD with respect to $\langle \widetilde{\mathcal{W}}, \tilde{b}, \prod_i G_i \rangle$.*

*Proof.* According to Definition 1, SBD-ness with respect to $\langle \widetilde{\mathcal{W}}, \tilde{b}, \prod_k G_k \rangle$ requires four properties:

1. There exists a group action $\cdot_\mathcal{Z} : G \times \mathcal{Z} \to \mathcal{Z}$,

2. Equivariance holds: $\forall g \in G, [w] \in \widetilde{\mathcal{W}}$, we have $g \cdot_{\mathcal{Z}} h \circ \widetilde{b}([w]) = h \circ \widetilde{b}(g \cdot_{\widetilde{\mathcal{W}}} [w])$,

3. There exists a decomposition $\mathcal{Z} = \mathcal{Z}_1 \times \cdots \times \mathcal{Z}_K$ and group actions $\cdot_k : G_k \times Z_k \to Z_k$ such that

$$(g_1, \ldots, g_K) \cdot_{\mathcal{Z}} (z_1, \ldots, z_K) = (g_1 \cdot_1 z_1, \ldots, g_K \cdot_K z_K),$$

4. $h$ is injective.

Properties 1, 3 and 4 are ensured by the fact that $h$ is SBD with respect to $\langle \mathcal{W}, b, \prod_k G_k \rangle$. We thus just need to prove the second property. Suppose $g \in G$ and $[w] \in \widetilde{\mathcal{W}}$, we have:

$$
\begin{aligned}
g \cdot_{\mathcal{Z}} h \circ \widetilde{b}([w]) &= g \cdot_{\mathcal{Z}} h \circ b(w) && \text{according to Definition 4} \\
&= h \circ b(g \cdot_{\mathcal{W}} w) && \text{since } h \text{ is SBD (equivariance property)} \\
&= h \circ \widetilde{b}([g \cdot_{\mathcal{W}} w]) && \text{according to Definition 4} \\
&= h \circ \widetilde{b}(g \cdot_{\widetilde{\mathcal{W}}} [w]) && \text{according to Definition 5}
\end{aligned}
$$

$\square$

**Proposition 8.** *Reciprocally, if $h$ is SBD with respect to $\langle \widetilde{\mathcal{W}}, \widetilde{b}, \prod_i G_i \rangle$, then $h$ is SBD with respect to $\langle \mathcal{W}, b, \prod_i G_i \rangle$.*

*Proof.* As in the previous proof, it suffices to establish equivariance in order to prove disentanglement with respect to $\langle \mathcal{W}, b, \prod_i G_i \rangle$. Let $g \in G$ and $w \in \mathcal{W}$ :

$$
\begin{aligned}
g \cdot_{\mathcal{Z}} h \circ b(w) &= g \cdot_{\mathcal{Z}} h \circ \widetilde{b}([w]) && \text{according to Definition 4} \\
&= h \circ \widetilde{b}(g \cdot_{\widetilde{\mathcal{W}}} [w]) && \text{since } h \text{ is SBD (equivariance property)} \\
&= h \circ \widetilde{b}([g \cdot_{\mathcal{W}} w]) && \text{according to Definition 5} \\
&= h \circ b(g \cdot_{\mathcal{W}} w) && \text{according to Definition 4}
\end{aligned}
$$

$\square$

We can now introduce the main result of this appendix:

**Theorem 1.** *If $h$ is SBD with respect to $\langle \mathcal{W}, b, \prod_k G_k \rangle$ then $\widetilde{b}$ is injective.*

*Proof.* Suppose $h$ is SBD with respect to $\langle \mathcal{W}, b, \prod_k G_k \rangle$. According to Proposition 7, $h$ is also SBD with respect to $\langle \widetilde{\mathcal{W}}, \widetilde{b}, \prod_k G_k \rangle$.

Let us assume that $\widetilde{b}$ is not injective and show that it leads to a contradiction.

Since $\widetilde{b}$ is not injective, there exist $[w_1] \neq [w_2]$ such that $\widetilde{b}([w_1]) = \widetilde{b}([w_2])$. Since $[w_1] \neq [w_2]$ then let $g \in G$ be the symmetry such that $b(g \cdot_{\mathcal{W}} w_1) \neq b(g \cdot_{\mathcal{W}} w_2)$. On the one hand:

$$
\begin{aligned}
b(g \cdot_{\mathcal{W}} w_1) \neq b(g \cdot_{\mathcal{W}} w_2) &\implies \widetilde{b}([g \cdot_{\mathcal{W}} w_1]) \neq \widetilde{b}([g \cdot_{\mathcal{W}} w_2]) && \text{according to Definition 2} \\
&\implies h \circ \widetilde{b}([g \cdot_{\mathcal{W}} w_1]) \neq h \circ \widetilde{b}([g \cdot_{\mathcal{W}} w_2]) && \text{since } h \text{ is injective} \\
&\implies g \cdot_{Z} h \circ \widetilde{b}([w_1]) \neq g \cdot_{Z} h \circ \widetilde{b}([w_2]) && \text{equivariance property}
\end{aligned}
$$

On the other hand, since $\widetilde{b}([w_1]) = \widetilde{b}([w_2])$, then $g \cdot_{Z} h \circ \widetilde{b}([w_1]) = g \cdot_{Z} h \circ \widetilde{b}([w_2])$. We have a contradiction and thus $\widetilde{b}$ is injective.

$\square$

From a prediction perspective this result is quite intuitive: for an injective encoder to allow accurate prediction of future observations $x'$ from $x$ and $g$, it is necessary for the observation to be unambiguous. From a disentanglement perspective this is a strong limitation, it means that all the features to disentangle have to be constantly observed by the agent.

We use this result to justify the introduction of Assumption 1 to derive our algorithm for the group decomposition. Since we have shown that LSB disentanglement is impossible when $\tilde{b}$ is not injective, it is a necessary assumption to assume.

## H    ADDITIONAL RESULTS

### H.1    PREDICTION BASED MODEL SELECTION

For unsupervised disentanglement learning, choosing the model with the best disentanglement without knowing the ground-truth features is a crucial and often difficult task. Several solutions have been discussed in the state of the art to adress this issue for purely unsupervised methods (Duan et al., 2019; Zhou et al., 2021; Holtz et al., 2022), they mostly rely on learning a batch of models with different hyperparameters or initialisation and selecting the ones sharing some defined properties. This model selection issue arises in our method at two different stages: the action clustering resulting of Step 2 and the representation learned by GMA-VAE.

As we are in a self-supervised framework, we investigate whether it is possible to select the model achieving the best disentanglement (measured here with the independence metric (Painter et al., 2020)) solely based on the prediction error. Similarly, we aim to determine whether it is possible to identify which A-VAE model results in the correct action clustering in Step 2. To evaluate clustering quality, we use the Adjusted Rand Index (ARI) (Hubert & Arabie, 1985), a metric to be maximized that equals 1 if and only if the predicted clustering exactly matches the ground-truth partition (up to a permutation).

For multiple hyperparameter configurations and random seeds, we plot for the COIL2 experiment the metric of interest as a function of the prediction error in Figure 10.

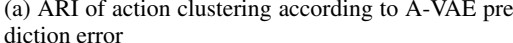

(a) ARI of action clustering according to A-VAE prediction error    (b) Independence according to prediction error

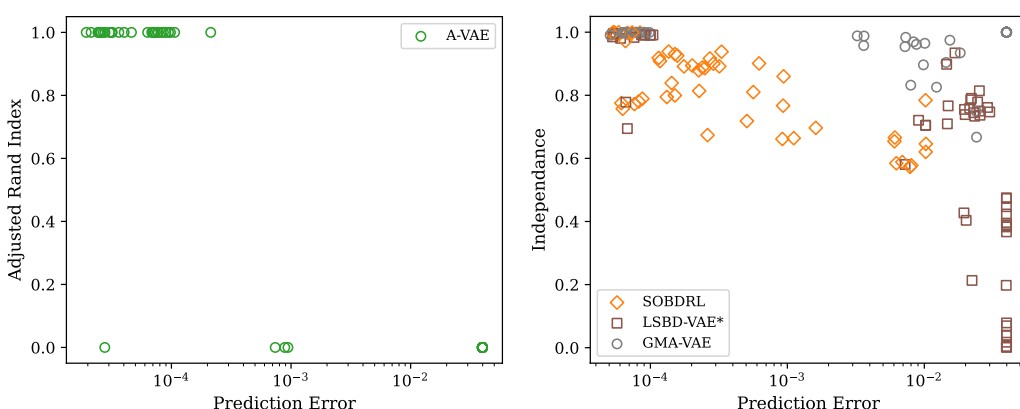

Figure 10: Metric of interest as a function of the prediction error

For A-VAE (left), we see that for almost all of the models, either the prediction performs well and the action clustering retrieves the correct partition, either the model cannot predict correctly and the action clustering retrieves a clustering only composed of singletons meaning that the ARI is equal to zero. However, few models out still has a good prediction error with a low ARI. It is therefore possible to train a batch of A-VAE models with different hyperparameters and initialisation seeds and take the action clustering that is retrieved by the majority of models achieving the best predictions.

For the disentangled representation learning algorithms (right), we can then notice that for the LSBD-VAE* and SOBDRL, they are few models that achieve a good prediction without being disentangled. However the model with the best prediction is always disentangled with a slight margin. For GMA-VAE, a good prediction always implies a disentanglement almost equal to 1.

## H.2 ASSUMPTION 2 FOR OTHER ALGORITHMS

This section aims to demonstrate that, although not explicitly stated, related LSBD algorithms implicitly rely on Assumption 2 (action disentanglement) to some extent.

First, Forward-VAE directly requires disentangled actions, as its action matrix parameterization is identical to GMA-VAE, with the vectors $\pi_k \in \{0, 1\}^d$ provided as prior knowledge.

To evaluate the extent to which SOBDRL and LSBD-VAE* depend on this assumption, we modify the COIL2 experimental setup by progressively relaxing Assumption 2. In this modified setting, each action corresponds to an element of the product of $k \geqslant 1$ distinct direct factors. We then measure the degree of disentanglement in the learned representation using the Independence score, as a function of the entanglement level $k$. The results, shown in Figure 11, indicate that the Independence score decreases as $k$ increases, and rapidly approaches the score obtained by A-VAE with fully entangled actions (represented by the green dotted line). These findings support the claim that Assumption 2, i.e., $k = 1$, is in fact necessary for these algorithms to consistently learn a disentangled representation.

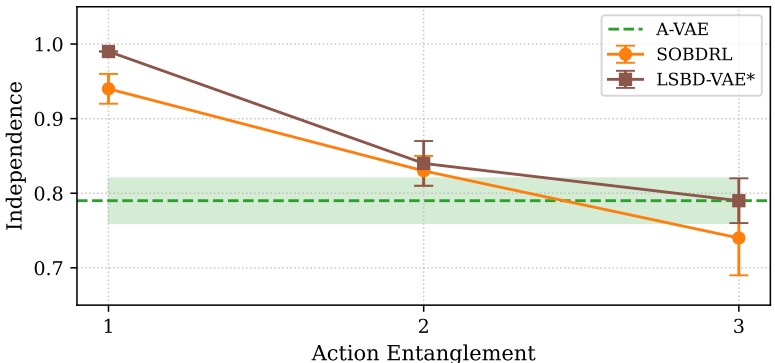

Figure 11: Disentanglement according to action entanglement

We also evaluated the disentanglement of those methods on the other metrics in Table 3. As we can see the disentanglement strictly decreases as actions are more entangled except for the MIG metric.

Table 3: Disentanglement with respect to action entanglement

|  | k | Beta-VAE | Inde | Mod | DCI | SAP | MIG |
|---|---|---|---|---|---|---|---|
| | 1 | $1.00 \pm .00$ | $.99 \pm .00$ | $1.00 \pm .00$ | $1.00 \pm .00$ | $.27 \pm .04$ | $.06 \pm .03$ |
| LSBD-VAE* | 2 | $.99 \pm .00$ | $.84 \pm .03$ | $.83 \pm .04$ | $.82 \pm .05$ | $.14 \pm .02$ | $.07 \pm .03$ |
| | 3 | $.78 \pm .05$ | $.69 \pm .03$ | $.59 \pm .05$ | $.62 \pm .05$ | $.03 \pm .01$ | $.04 \pm .01$ |
| | 1 | $.99 \pm .01$ | $.94 \pm .02$ | $.94 \pm .04$ | $.88 \pm .03$ | $.39 \pm .04$ | $.20 \pm .07$ |
| SOBDRL | 2 | $.94 \pm .03$ | $.83 \pm .02$ | $.81 \pm .02$ | $.70 \pm .06$ | $.30 \pm .06$ | $.04 \pm .03$ |
| | 3 | $.74 \pm .09$ | $.74 \pm .05$ | $.68 \pm .05$ | $.48 \pm .05$ | $.23 \pm .06$ | $.19 \pm .05$ |

## H.3 Latent Manipulation

Here we aim to show that the learned disentangled representation allow for latent manipulation. In the discrete case, we select two images of the 3DShapes dataset that we encode with GMA-VAE, then we construct several latent representations by mixing the different latent factors of the two images. Finally, we decode those new latent representations to obtain new images. The results are shown in Figure 12. We can see that by mixing the different latent factors, we can control the different features of the decoded images.

For the continuous case, we select one image of MPI3D that we encode with GMA-VAE. For each feature (vertical angle and horizontal angle), we successively apply a rotation of a fixed angle to the corresponding latent factor while keeping the other latent factor unchanged. The decoded images are shown in Figure 13. We can see that by rotating the latent factors, we can control the rotation of the object in the decoded images.

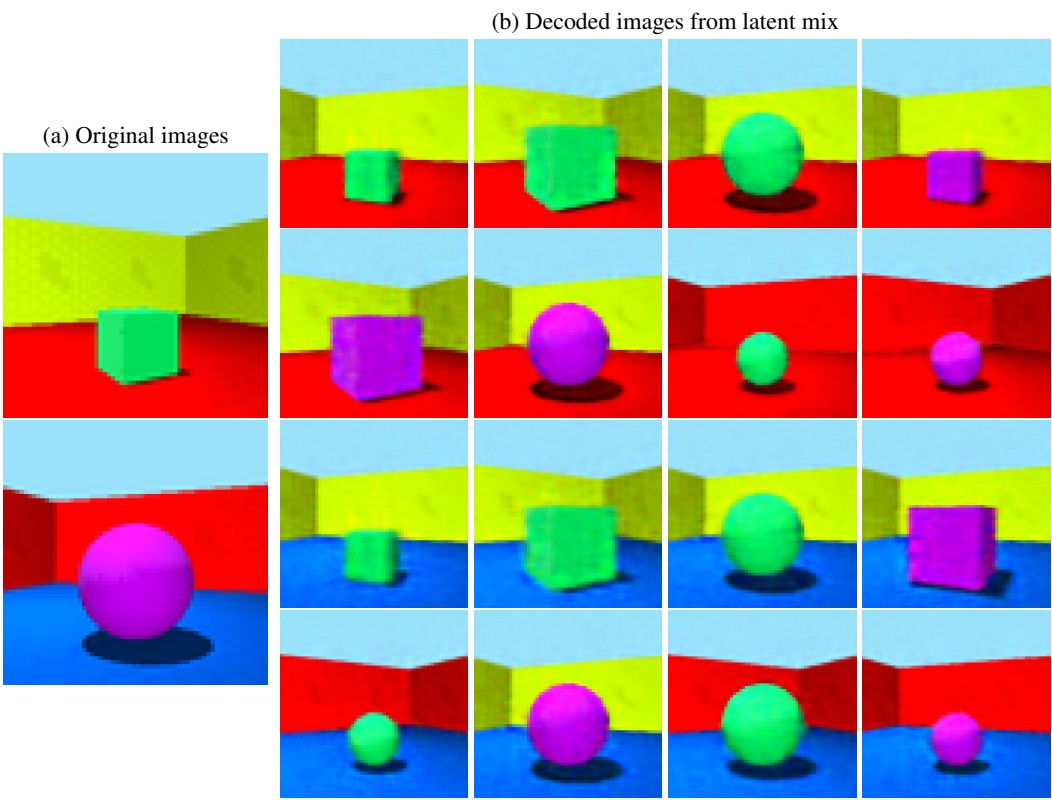

Figure 12: Decoding mixed latent representations of 3DShapes images

## H.4 Disentanglement metrics

We show all the disentanglement and prediction error results of the different methods on the different datasets in Table 4 to 14, the values reported correspond to the mean over five seeds with the 68% confidence interval. The random method corresponds to a randomly initialized neural network that was not trained.

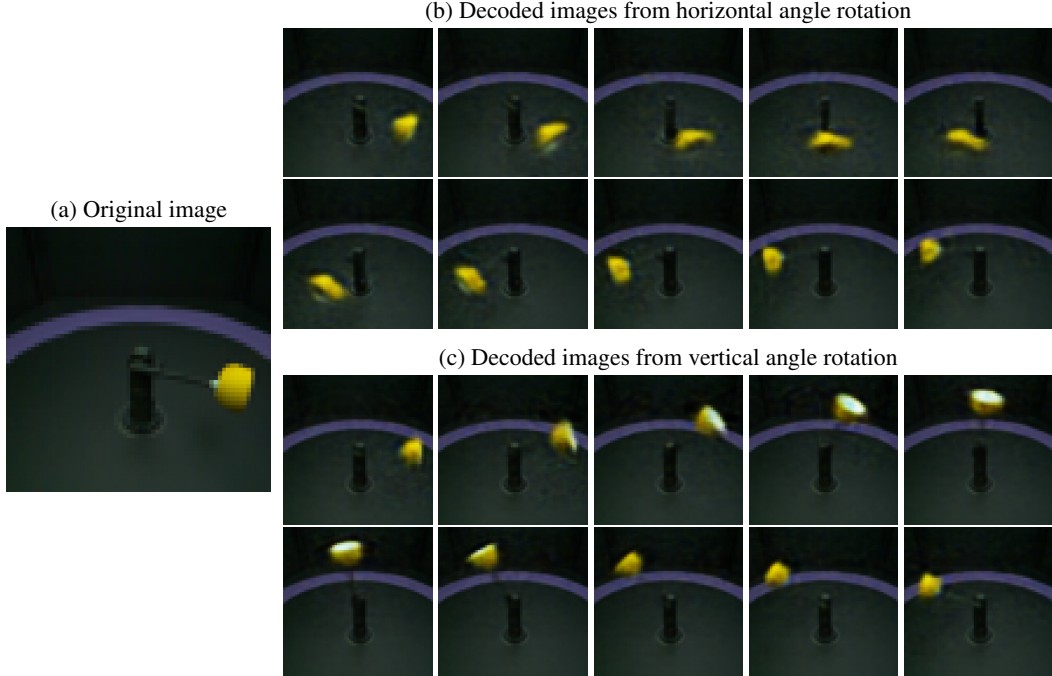

(a) Original image

(b) Decoded images from horizontal angle rotation

(c) Decoded images from vertical angle rotation

Figure 13: Decoding rotated latent representations of MPI3D image

Table 4: Disentanglement for FlatLand with rotation colors

|  |  | ↑ Beta-VAE | ↑ Inde | ↑ Mod | ↑ DCI | ↑ SAP | ↑ MIG | ↓ Prediction |
|---|---|---|---|---|---|---|---|---|
| LSBD-VAE | mean | **1.00 ± .00** | .96 ± .00 | **1.00 ± .00** | .98 ± .01 | **.53 ± .05** | .10 ± .04 | 1.6e-09 ± 4.8e-10 |
|  | best | 1.00 | .96 | 1.00 | 1.00 | .46 | .23 | 4.8e-10 |
| LSBD-VAE* | mean | **1.00 ± .00** | .93 ± .02 | .90 ± .06 | .87 ± .07 | .40 ± .03 | .07 ± .03 | 9.6e-10 ± 5.3e-10 |
|  | best | 1.00 | .97 | 1.00 | .98 | .53 | .02 | 3.3e-9 |
| SOBDRL | mean | .93 ± .03 | .78 ± .02 | .85 ± .04 | .53 ± .09 | .28 ± .04 | .10 ± .02 | 2.7e-5 ± 2.4e-5 |
|  | best | .99 | .84 | .77 | .56 | .44 | .02 | 1.6e-8 |
| GMA-VAE | mean | **1.00 ± .00** | **1.00 ± .00** | **1.00 ± .00** | **1.00 ± .00** | **.54 ± .04** | .01 ± .00 | 1.6e-10 ± 2.9e-11 |
|  | best | 1.00 | 1.00 | 1.00 | 1.00 | .51 | .00 | 1.3e-10 |
| A-VAE | mean | .89 ± .05 | .84 ± .01 | .76 ± .01 | .44 ± .03 | .11 ± .02 | .05 ± .01 | **6.6e-11 ± 1.7e-11** |
|  | best | .95 | .82 | .77 | .48 | .03 | .05 | 3.0e-11 |
| β-VAE |  | .68 ± .06 | .77 ± .01 | .88 ± .03 | .22 ± .05 | .20 ± .04 | .13 ± .03 |  |
| Factor-VAE |  | .69 ± .07 | .69 ± .04 | .81 ± .03 | .24 ± .06 | .29 ± .06 | **.16 ± .05** |  |
| DIP-VAE I |  | .56 ± .03 | .67 ± .01 | .78 ± .01 | .12 ± .02 | .11 ± .01 | .04 ± .01 |  |
| DIP-VAE II |  | .84 ± .04 | .62 ± .02 | .74 ± .00 | .17 ± .01 | .18 ± .04 | .05 ± .01 |  |
| Random |  | .41 ± .03 | .51 ± .01 | .61 ± .04 | .05 ± .01 | .05 ± .00 | .04 ± .01 |  |

Table 5: Disentanglement for FlatLand with permutation colors

|  |  | ↑ Beta-VAE | ↑ Inde | ↑ Mod | ↑ DCI | ↑ SAP | ↑ MIG | ↓ Prediction |
|---|---|---|---|---|---|---|---|---|
| LSBD-VAE | mean | **1.00 ± .00** | .97 ± .00 | **1.00 ± .00** | .95 ± .02 | .46 ± .04 | .04 ± .02 | 4.1e-6 ± 7.3e-7 |
|  | best | 1.00 | .97 | 1.00 | .98 | .59 | .02 | 2.6e-6 |
| LSBD-VAE* | mean | **1.00 ± .00** | .98 ± .00 | **1.00 ± .00** | .99 ± .01 | .40 ± .06 | **.08 ± .02** | 3.3e-6 ± 2.8e-7 |
|  | best | 1.00 | .98 | 1.00 | 1.00 | .46 | .00 | 2.7e-6 |
| SOBDRL | mean | **1.00 ± .00** | .74 ± .01 | .79 ± .01 | .50 ± .02 | .25 ± .03 | **.08 ± .02** | 3.6e-3 ± 8.3e-7 |
|  | best | 1.00 | .76 | .80 | .43 | .16 | .12 | 3.6e-3 |
| GMA-VAE | mean | **1.00 ± .00** | .99 ± .00 | **1.00 ± .00** | .99 ± .01 | **.51 ± .04** | .08 ± .01 | 5.8e-6 ± 3.1e-6 |
|  | best | 1.00 | 1.00 | 1.00 | 1.00 | .59 | .11 | 1.0e-6 |
| A-VAE | mean | .87 ± .06 | .89 ± .04 | .92 ± .04 | .47 ± .08 | .15 ± .05 | **.09 ± .01** | **1.1e-6 ± 1.9e-7** |
|  | best | 1.00 | .95 | 1.00 | .47 | .28 | .07 | 1.0e-6 |
| β-VAE |  | .64 ± .03 | .60 ± .03 | .80 ± .02 | .13 ± .04 | .12 ± .02 | .03 ± .01 |  |
| Factor-VAE |  | .77 ± .05 | .61 ± .01 | .79 ± .02 | .13 ± .02 | .17 ± .04 | .05 ± .02 |  |
| DIP-VAE I |  | .60 ± .06 | .71 ± .02 | .87 ± .02 | .11 ± .03 | .10 ± .04 | .06 ± .02 |  |
| DIP-VAE II |  | .70 ± .05 | .71 ± .01 | .81 ± .01 | .11 ± .02 | .10 ± .02 | .04 ± .01 |  |
| Random |  | .46 ± .01 | .59 ± .01 | .68 ± .02 | .05 ± .01 | .09 ± .01 | .03 ± .01 |  |

Table 6: Disentanglement for COIL2

| | | ↑ Beta-VAE | ↑ Inde | ↑ Mod | ↑ DCI | ↑ SAP | ↑ MIG | ↓ Prediction |
|---|---|---|---|---|---|---|---|---|
| LSBD-VAE | mean | **1.00 ± .00** | .99 ± .00 | **1.00 ± .00** | .98 ± .01 | .31 ± .05 | .01 ± .01 | 8.2e-5 ± 3.0e-6 |
| | best | 1.00 | .99 | 1.00 | .92 | .43 | .00 | 7.4e-5 |
| LSBD-VAE* | mean | **1.00 ± .00** | .99 ± .00 | **1.00 ± .00** | .97 ± .02 | .25 ± .05 | .04 ± .03 | 9.4e-5 ± 3.1e-6 |
| | best | 1.00 | .99 | .98 | .91 | .45 | .00 | 8.9e-5 |
| SOBDRL | mean | .99 ± .00 | .91 ± .05 | .90 ± .05 | .90 ± .05 | .52 ± .05 | .31 ± .06 | **7.2e-5 ± 5.0e-6** |
| | best | 1.00 | 1.00 | .98 | 1.00 | .60 | .42 | 5.4e-5 |
| GMA-VAE | mean | **1.00 ± .00** | **1.00 ± .00** | **1.00 ± .00** | **1.00 ± .00** | **.64 ± .04** | **.46 ± .02** | 7.4e-5 ± 5.5e-6 |
| | best | 1.00 | 1.00 | 1.00 | 1.00 | .71 | .48 | 6.7e-5 |
| A-VAE | mean | .79 ± .03 | .79 ± .03 | .67 ± .05 | .58 ± .03 | .08 ± .02 | .09 ± .03 | 1.0e-4 ± 2.5e-5 |
| | best | .81 | .85 | .75 | .55 | .04 | .05 | 7.0e-5 |
| β-VAE | | .67 ± .03 | .75 ± .01 | .83 ± .01 | .36 ± .01 | .30 ± .01 | .24 ± .04 | |
| Factor-VAE | | .81 ± .09 | .78 ± .01 | .87 ± .01 | .50 ± .05 | .34 ± .02 | .23 ± .02 | |
| DIP-VAE I | | .74 ± .01 | .71 ± .01 | .83 ± .02 | .41 ± .02 | .32 ± .02 | .22 ± .03 | |
| DIP-VAE II | | .84 ± .02 | .75 ± .01 | .83 ± .03 | .42 ± .01 | .37 ± .04 | .16 ± .02 | |
| Random | | .58 ± .06 | .64 ± .02 | .83 ± .02 | .27 ± .03 | .11 ± .02 | .13 ± .02 | |

Table 7: Disentanglement for COIL3

| | | ↑ Beta-VAE | ↑ Inde | ↑ Mod | ↑ DCI | ↑ SAP | ↑ MIG | ↓ Prediction |
|---|---|---|---|---|---|---|---|---|
| LSBD-VAE | mean | **1.00 ± .00** | .98 ± .00 | **1.00 ± .00** | .97 ± .01 | **.42 ± .04** | .08 ± .02 | **1.2e-4 ± 5.4e-6** |
| | best | 1.00 | .98 | 1.00 | .94 | .40 | .12 | 1.0e-04 |
| LSBD-VAE* | mean | .86 ± .06 | .92 ± .02 | .83 ± .03 | .75 ± .03 | .21 ± .04 | .12 ± .02 | 3.2e-3 ± 6.2e-4 |
| | best | 1.00 | .96 | .88 | .86 | .28 | .13 | 2.6e-04 |
| SOBDRL | mean | .86 ± .09 | .88 ± .03 | .82 ± .05 | .66 ± .07 | .26 ± .08 | .10 ± .02 | 4.7e-4 ± 1.3e-4 |
| | best | .96 | .91 | .87 | .80 | .42 | .12 | 2.6e-04 |
| GMA-VAE | mean | **1.00 ± .00** | **1.00 ± .00** | **1.00 ± .00** | **1.00 ± .00** | .35 ± .03 | **.17 ± .02** | **1.1e-4 ± 3.7e-6** |
| | best | 1.00 | 1.00 | 1.00 | 1.00 | .34 | .13 | 9.9e-05 |
| A-VAE | mean | .49 ± .01 | .80 ± .02 | .61 ± .01 | .15 ± .01 | .03 ± .01 | .04 ± .01 | 2.8e-4 ± 1.9e-5 |
| | best | .50 | .85 | .62 | .13 | .02 | .04 | 2.0e-04 |
| β-VAE | | .44 ± .05 | .82 ± .01 | .76 ± .02 | .10 ± .03 | .04 ± .01 | .08 ± .01 | |
| Factor-VAE | | .74 ± .02 | .84 ± .01 | .80 ± .01 | .28 ± .02 | .11 ± .02 | .13 ± .03 | |
| DIP-VAE I | | .41 ± .01 | .84 ± .01 | .78 ± .01 | .04 ± .00 | .06 ± .01 | .04 ± .00 | |
| DIP-VAE II | | .49 ± .05 | .82 ± .01 | .75 ± .01 | .10 ± .01 | .08 ± .02 | .06 ± .01 | |
| Random | | .38 ± .02 | .65 ± .01 | .77 ± .01 | .05 ± .01 | .08 ± .02 | .05 ± .01 | |

Table 8: Disentanglement for 3DShapes

| | | ↑ Beta-VAE | ↑ Inde | ↑ Mod | ↑ DCI | ↑ SAP | ↑ MIG | ↓ Prediction |
|---|---|---|---|---|---|---|---|---|
| LSBD-VAE | mean | **.98 ± .00** | .98 ± .00 | **1.00 ± .00** | .99 ± .00 | .43 ± .03 | .07 ± .01 | 7.6e-4 ± 2.2e-5 |
| | best | .98 | .99 | 1.00 | 1.00 | .45 | .04 | 6.8e-4 |
| LSBD-VAE* | mean | .97 ± .00 | .97 ± .01 | **1.00 ± .00** | .95 ± .02 | .49 ± .04 | .09 ± .02 | 1.1e-3 ± 2.6e-4 |
| | best | .97 | .98 | 1.00 | .98 | .59 | .03 | 8.3e-4 |
| SOBDRL | mean | .91 ± .05 | .93 ± .02 | .96 ± .03 | .85 ± .07 | .41 ± .05 | **.22 ± .05** | 2.0e-3 ± 8.7e-4 |
| | best | .98 | .97 | 1.00 | .97 | .31 | .26 | 9.2e-4 |
| GMA-VAE | mean | **.98 ± .00** | **1.00 ± .00** | **1.00 ± .00** | **1.00 ± .00** | **.56 ± .04** | **.26 ± .02** | **6.0e-4 ± 3.2e-5** |
| | best | .98 | 1.00 | 1.00 | 1.00 | .69 | .19 | 5.1e-4 |
| A-VAE | mean | .35 ± .03 | .88 ± .01 | .75 ± .00 | .15 ± .02 | .06 ± .01 | .04 ± .01 | 8.1e-4 ± 3.6e-5 |
| | best | .42 | .90 | .74 | .18 | .07 | .02 | 8.2e-4 |
| β-VAE | | .52 ± .04 | .72 ± .01 | .74 ± .00 | .22 ± .04 | .14 ± .01 | .08 ± .01 | |
| Factor-VAE | | .63 ± .03 | .74 ± .01 | .76 ± .01 | .35 ± .01 | .20 ± .05 | .11 ± .03 | |
| DIP-VAE I | | .62 ± .02 | .71 ± .02 | .75 ± .00 | .28 ± .02 | .15 ± .01 | .12 ± .02 | |
| DIP-VAE II | | .67 ± .03 | .69 ± .01 | .75 ± .01 | .37 ± .01 | .14 ± .01 | .22 ± .02 | |
| Random | | .31 ± .04 | .64 ± .01 | .70 ± .01 | .04 ± .00 | .04 ± .00 | .04 ± .00 | |

Table 9: Disentanglement for MPI3D

| | | ↑ Beta-VAE | ↑ Inde | ↑ Mod | ↑ DCI | ↑ SAP | ↑ MIG | ↓ Prediction |
|---|---|---|---|---|---|---|---|---|
| SOBDRL | mean | **1.00 ± .00** | .93 ± .01 | **.47 ± .07** | .32 ± .02 | **.28 ± .05** | **.04 ± .02** | 8.5e-4 ± 5.9e-5 |
| | best | 1.00 | .93 | .40 | .29 | .14 | .02 | 6.8e-4 |
| HAE | mean | **1.00 ± .00** | .71 ± .06 | **.42 ± .07** | .19 ± .05 | **.29 ± .07** | .02 ± .00 | **4.6e-4 ± 2.4e-5** |
| | best | 1.00 | .96 | .40 | .39 | .54 | 2.6e-3 | 4.7e-4 |
| GMA-VAE | mean | **1.00 ± .00** | **.97 ± .01** | **.49 ± .08** | **.53 ± .04** | .21 ± .04 | **.04 ± .01** | 1.6e-3 ± 5.8e-4 |
| | best | 1.00 | .99 | .40 | .58 | .15 | .06 | 3.5e-4 |
| β-VAE | | .81 ± .03 | .53 ± .01 | **.41 ± .06** | .02 ± .00 | .08 ± .01 | .03 ± .00 | |
| Random | | .63 ± .03 | .52 ± .01 | .36 ± .05 | 8.4e-3 ± 2.6e-3 | .06 ± .02 | .02 ± .00 | |

Table 10: Disentanglement for noisy MPI3D

|  |  | ↑ Beta-VAE | ↑ Inde | ↑ Mod | ↑ DCI | ↑ SAP | ↑ MIG | ↓ Prediction |
|---|---|---|---|---|---|---|---|---|
| SOBDRL | mean | .99 ± .00 | .79 ± .05 | **.48 ± .07** | .20 ± .05 | .41 ± .07 | **.04 ± .01** | 2.8e-3 ± 2.4e-4 |
|  | best | 1.00 | .90 | .79 | .26 | .45 | .01 | 3.3e-3 |
| HAE | mean | **1.00 ± .00** | .79 ± .06 | **.47 ± .05** | .22 ± .05 | .32 ± .09 | **.03 ± .01** | **1.7e-3 ± 3.1e-5** |
|  | best | 1.00 | .90 | .40 | .27 | .45 | .01 | 1.7e-3 |
| GMA-VAE | mean | **1.00 ± .00** | **.90 ± .00** | **.50 ± .08** | **.31 ± .02** | **.51 ± .04** | **.04 ± .01** | **1.6e-3 ± 3.6e-5** |
|  | best | 1.00 | .90 | .84 | .31 | .64 | .03 | 1.6e-3 |
| β-VAE |  | .81 ± .03 | .53 ± .01 | .41 ± .06 | .02 ± .00 | .08 ± .01 | **.03 ± .00** |  |
| Random |  | .63 ± .03 | .52 ± .01 | .36 ± .05 | 8.4e-3 ± 2.6e-3 | .06 ± .02 | .02 ± .00 |  |

Table 11: Disentanglement for COIL2 in iid setting

|  | ↑ Beta-VAE | ↑ Inde | ↑ Mod | ↑ DCI | ↑ SAP | ↑ MIG |
|---|---|---|---|---|---|---|
| LSBDVAE | **1.00 ± .00** | .99 ± .00 | .99 ± .00 | **.95 ± .02** | .31 ± .06 | .05 ± .03 |
| LSBD-VAE* | **1.00 ± .00** | .99 ± .00 | **1.00 ± .00** | **.97 ± .01** | .28 ± .05 | .09 ± .03 |
| SOBDRL | .97 ± .02 | .90 ± .05 | .94 ± .04 | .88 ± .05 | **.44 ± .05** | .27 ± .05 |
| GMA-VAE | **1.00 ± .00** | **1.00 ± .00** | **1.00 ± .00** | **.98 ± .01** | **.48 ± .04** | **.47 ± .01** |
| A-VAE | .76 ± .05 | .75 ± .02 | .71 ± .05 | .53 ± .03 | .06 ± .02 | .11 ± .02 |

Table 12: Disentanglement for COIL2 in ood setting

|  | ↑ Beta-VAE | ↑ Inde | ↑ Mod | ↑ DCI | ↑ SAP | ↑ MIG |
|---|---|---|---|---|---|---|
| LSBDVAE | **1.00 ± .00** | .99 ± .00 | **1.00 ± .00** | .98 ± .01 | .26 ± .05 | .04 ± .03 |
| LSBD-VAE* | **1.00 ± .00** | .94 ± .04 | .98 ± .02 | **.95 ± .04** | .34 ± .03 | .10 ± .06 |
| SOBDRL | **1.00 ± .00** | .88 ± .03 | .97 ± .01 | .92 ± .02 | **.53 ± .05** | .18 ± .07 |
| GMA-VAE | .99 ± .00 | **1.00 ± .00** | **1.00 ± .00** | .98 ± .02 | **.47 ± .07** | **.39 ± .09** |
| A-VAE | .67 ± .03 | .76 ± .01 | .92 ± .02 | .53 ± .03 | .08 ± .02 | .06 ± .02 |

Table 13: Disentanglement for COIL3 in iid setting

|  | ↑ Beta-VAE | ↑ Inde | ↑ Mod | ↑ DCI | ↑ SAP | ↑ MIG |
|---|---|---|---|---|---|---|
| LSBDVAE | **1.00 ± .00** | .98 ± .00 | **1.00 ± .00** | .97 ± .01 | .38 ± .02 | .09 ± .01 |
| LSBD-VAE* | .66 ± .07 | .86 ± .01 | .86 ± .01 | .28 ± .08 | .18 ± .04 | .11 ± .02 |
| SOBDRL | .98 ± .01 | .93 ± .01 | .91 ± .02 | .76 ± .05 | **.36 ± .05** | **.16 ± .04** |
| GMA-VAE | **1.00 ± .00** | **1.00 ± .00** | **1.00 ± .00** | **1.00 ± .00** | .23 ± .04 | **.17 ± .03** |
| A-VAE | .48 ± .02 | .80 ± .01 | .61 ± .01 | .15 ± .02 | .01 ± .00 | .02 ± .00 |

Table 14: Disentanglement for COIL3 in ood setting

|  | ↑ Beta-VAE | ↑ Inde | ↑ Mod | ↑ DCI | ↑ SAP | ↑ MIG |
|---|---|---|---|---|---|---|
| LSBDVAE | **1.00 ± .00** | **.99 ± .00** | 1.00 ± .00 | **.99 ± .00** | **.43 ± .04** | .05 ± .02 |
| LSBD-VAE* | .71 ± .09 | .85 ± .02 | .89 ± .03 | .38 ± .12 | .20 ± .03 | .07 ± .03 |
| SOBDRL | .97 ± .02 | .89 ± .03 | .97 ± .01 | .82 ± .06 | .33 ± .03 | .12 ± .02 |
| GMA-VAE | **1.00 ± .00** | **.99 ± .00** | **1.00 ± .00** | **.99 ± .00** | .29 ± .03 | **.18 ± .03** |
| A-VAE | .55 ± .03 | .74 ± .03 | .78 ± .02 | .18 ± .01 | .04 ± .01 | .04 ± .01 |

## H.5 $d_G$ MATRICES

Figure 14 presents the $d_G$ matrices for five different environments. Each matrix is normalized such that the clustering threshold corresponds to a distance of 1. Each black square represents available actions $\mathcal{G}_k$ associated with specific subgroups $G_k$. These blocks have pairwise distances strictly below 1, whereas the distances between two actions belonging to different subgroups are strictly greater than 1, resulting in a correct recovering of the ground-truth action partition.

## I  IMPLEMENTATION DETAILS

### I.1  DATASETS

Here, we describe the datasets used in this paper. A visual representation of the observation and the available actions are illustrated in Figure 15.

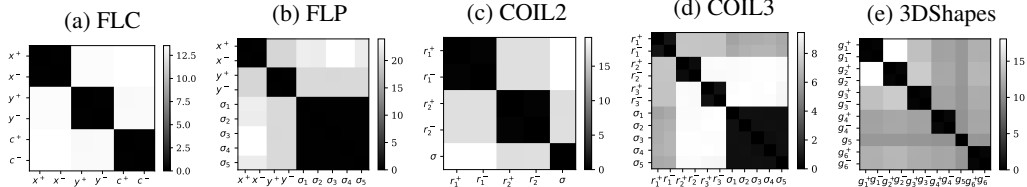

Figure 14: $d_G$ matrices averaged over 5 seeds, normalized so that the clustering threshold is 1.

Our implementation of Colored Flatland (Caselles-Dupré et al., 2018) consists of RGB images of size $64 \times 64 \times 3$ with a ball of radius 17 pixels. The ball can occupy 5 distinct positions along each axis, corresponding to $G_X = G_Y = \mathbb{Z}/5\mathbb{Z}$. In the cyclic color experiment (FLC), the base color is $[1, 0, 0]$, and the two available actions in $\mathcal{G}_C$ perform cyclic shifts of the active color channel in one direction. In the permutation color experiment (FLP), the base color is $[1/3, 2/3, 1]$, and each available action corresponds to a permutation of the RGB channels.

Our implementation of COIL (Nene et al., 1996) consists of RGB images of size $64 \times 64n \times 3$, where $n$ is the number of objects present in the scene. The group actions, and consequently the number of possible orientations for each object, are specified in Table 15. In this table we denote by $\sigma$ a permutation that permutes all objects, and by $r_i$ the rotation by one unit angle of the $i$-th object. The first two rows of Table 15 correspond to the datasets COIL2 and COIL3, which are used in the most of the experiments. The remaining rows describe additional environments used for action clustering experiments in Section 5.2.

Our implementation of 3DShapes (Burgess & Kim, 2018) consists of RGB images of size $64 \times 64 \times 3$, showing a 3D object placed at the center of a colored room. Each original generative factor was sub-sampled by a factor of 2, resulting in the following cardinalities: wall hue: 5, object hue; 5, background hue: 5, object scale: 4, object shape: 2, and viewing angle: 7. For each generative factor $i$, we define a cyclic symmetry group $G_i = \mathbb{Z}/k_i\mathbb{Z}$ over its possible values, along with two available actions $\mathcal{G}_i = \{g_i^-, g_i^+\}$ corresponding to unit rotation in each direction.

Our implementation of MPI3D (Gondal et al., 2019) consists of RGB images of size $64 \times 64 \times 3$, showing a robotic arm manipulating a colored object. Among the original generative factors, we only kept on two of them: horizontal and vertical object rotation angles, each having 40 different values. In order to approximate continuous symmetry groups, the actions are represented by $g = (j, \theta)$ with $j \in \{0, 1\}$ the axis of rotation and $\theta \in ]0, 2\pi[$ the rotation angle. In the noisy version of MPI3D, we add a Gaussian noise of standard deviation $2\pi/15$ on the rotation angle after applying the action.

Table 15: COIL Environments used

| $G$ | $\mathcal{G}_{\mathfrak{S}}$ | $\mathcal{G}_1$ | $\mathcal{G}_2$ | $\mathcal{G}_3$ | $\mathcal{G}_4$ | includes $e$ ? |
|---|---|---|---|---|---|---|
| $\mathfrak{S}_2 \times \frac{\mathbb{Z}}{7\mathbb{Z}} \times \frac{\mathbb{Z}}{5\mathbb{Z}}$ | $G^*_{\mathfrak{S}}$ | $\{-r_1, r_1\}$ | $\{-r_2, r_2\}$ | | | No |
| $\mathfrak{S}_3 \times \frac{\mathbb{Z}}{5\mathbb{Z}} \times \frac{\mathbb{Z}}{5\mathbb{Z}} \times \frac{\mathbb{Z}}{3\mathbb{Z}}$ | $G^*_{\mathfrak{S}}$ | $\{-r_1, r_1\}$ | $\{-r_2, r_2\}$ | $\{-r_3, r_3\}$ | | No |
| $\mathfrak{S}_3 \times \frac{\mathbb{Z}}{7\mathbb{Z}} \times \frac{\mathbb{Z}}{5\mathbb{Z}} \times \frac{\mathbb{Z}}{3\mathbb{Z}}$ | $\{\sigma\}$ | $\{r_1, 3r_1\}$ | $\{3r_2, 4r_2\}$ | $\{r_3\}$ | | Yes |
| $\mathfrak{S}_3 \times \frac{\mathbb{Z}}{3\mathbb{Z}} \times \frac{\mathbb{Z}}{7\mathbb{Z}} \times \frac{\mathbb{Z}}{3\mathbb{Z}}$ | $\{\sigma\}$ | $\{2r_1, 6r_1\}$ | $\{2r_2\}$ | $\{r_3\}$ | | Yes |
| $\mathfrak{S}_4 \times \frac{\mathbb{Z}}{7\mathbb{Z}} \times \frac{\mathbb{Z}}{5\mathbb{Z}} \times \frac{\mathbb{Z}}{3\mathbb{Z}} \times \frac{\mathbb{Z}}{3\mathbb{Z}}$ | $\{\sigma, \sigma^{-1}\}$ | $\{r_1, 3r_1\}$ | $\{3r_2, 4r_2\}$ | $\{r_3\}$ | $\{r_4\}$ | No |

## I.2 NETWORK ARCHITECTURES AND HYPERPARAMETERS

For all the experiments, dataset and method, we used Adam (Kingma & Ba, 2015) with the default parameters given by pytorch, a batch size of 16 and the auto-encoder architecture given Table 16.

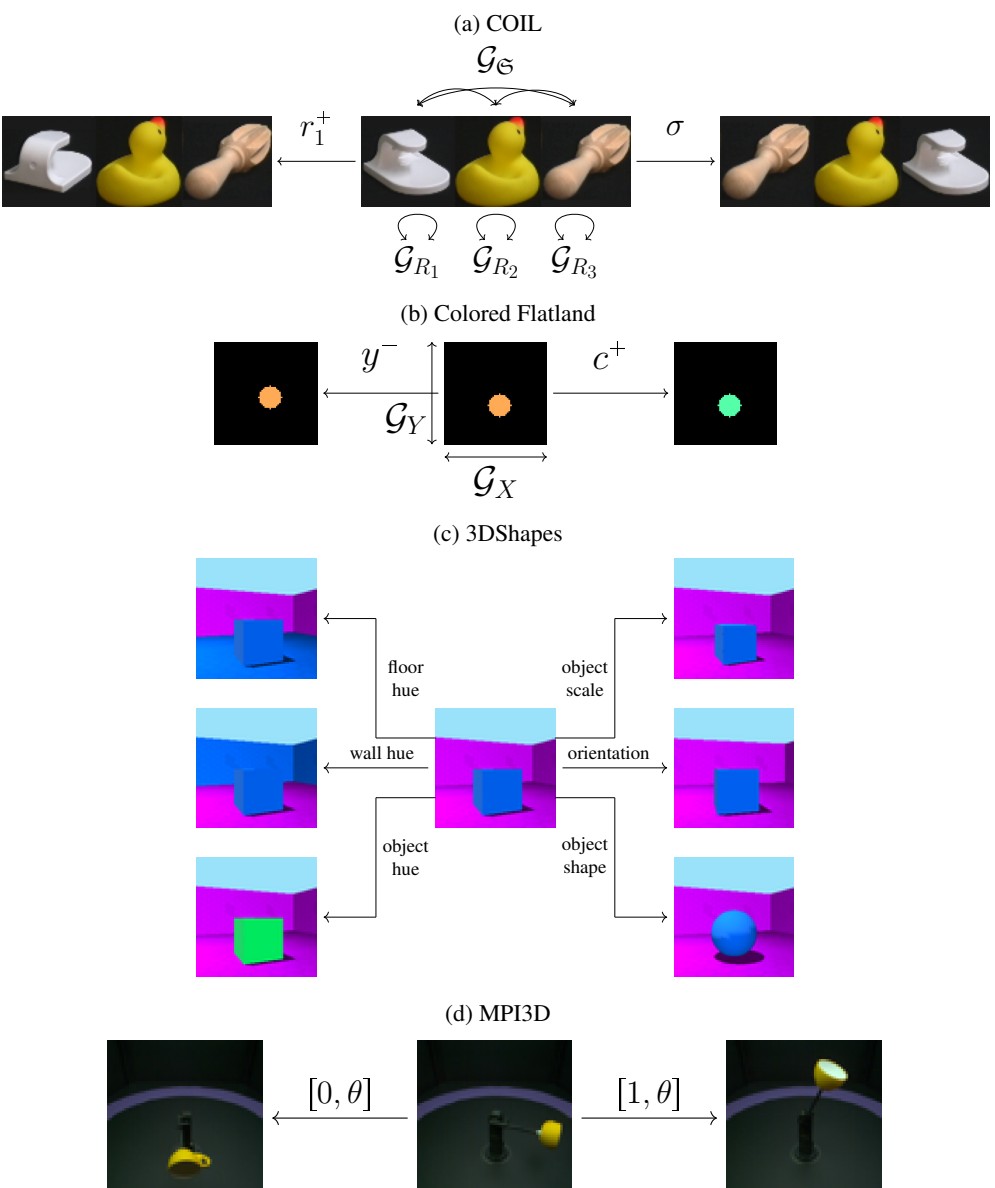

Figure 15: Presentation of the environments

Table 16: Auto-encoder architecture used for all methods and all datasets

| ENCODER | |
|---|---|
| Input | Size $(x,\ y,\ c)$ |
| Conv | Channels: 32, Kernel size: 8, Stride: 4, Padding: 2, ReLU |
| Conv | Channels: 64, Kernel size: 8, Stride: 4, Padding: 2, ReLU |
| Reshape | Flatten into $x/4 \times y/4 \times 64$ |
| Dense | Dimension: 256, ReLU |
| Dense | Dimension: depends on $d$ and the noise parametrisation |
| DECODER | |
| Input | Size $d$ |
| Dense | Dimension: 256, ReLU |
| Dense | Dimension: $x/4 \times y/4 \times 64$, ReLU |
| Reshape | $(x/4,\ y/4,\ 64)$ |
| ConvT | Channels: 32, Kernel size: 8, Stride: 4, Padding: 2, ReLU |
| ConvT | Channels: $c$, Kernel size: 8, Stride: 4, Padding: 2, Sigmoid |

For discrete action, an action matrix $A_g$ of SOBDRL is parametrized by $d(d-1)$ scalars $\theta_{i,j}^g$, one angle for each pair of plane $(i,j)$, the action matrix is then constructed by multiplying all the rotations matrices $A_g = \prod_{i<j} R_{i,j}(\theta_{i,j}^g)$. For LSBD-VAE, the block-diagonal actions matrices are already given as it is a supervised method. For LSBD-VAE* the only difference is that each block is learned with a $SO$ parametrisation similarly to SOBDRL. For our method, the action matrices are parametrized directly with the matrices coefficient, we used a tanH activation to ensure stability.

For continuous action, each action representation are neural networks of two hidden layer of 256 units with ReLU activation functions. SOBDRL action representation has an output of size $d(d-1)$ and use the same $SO$ parametrisation as for discrete action to construct the action matrix. GMA-VAE action representation has an output of size $d \times d$ and directly parametrize the action matrix coefficients. HAE action representation has as output a block-diagonal matrix of size $d \times d$ with the block shape given as prior knowledge, a matrix exponential is used on each block.

SOBDRL and HAE require multi-steps trajectories $(x_t, g_t, \ldots, x_{t+T-1}, g_{t+T-1}, x_{t+T})$ with $T > 1$. To have a fair comparison, every experiment will use sequences with $T = 5$, the other methods process independently each transition $(x_{t+k}, g_{t+k}, x_{t+k+1})$.

For A-VAE and GMA-VAE, we used a fixed latent noise of standard deviation $\sigma = 0.1$. For the action clustering of Step 2 we used the threshold $\eta = \sigma$ and $M = 2$. For the GMA-VAE masking, we force each sub-group to have at least one dimension assigned to it.

All hyperparameter details for each experiment are provided in the `config` folder of the GitHub repository.

## J    USE OF LARGE LANGUAGE MODELS

LLMs were moderately used to help in literature reviewing and English writing.

