# OpenReview forum: "Disentangled representation learning through unsupervised symmetry group discovery"
_ICLR.cc/2026/Conference — ICLR 2026 Poster_

### Official Review · Reviewer_GrsJ · 2025-10-29

**Soundness:** 3
**Presentation:** 3
**Contribution:** 3
**Rating:** 8
**Confidence:** 4

**Summary:**

This paper introduces a method to learn disentangled representations using symmetry groups without requiring prior knowledge of the group structure. The approach consists of three stages: learning an entangled representation (A-VAE), discovering the group decomposition via clustering, and learning a disentangled representation (GMA-VAE). This addresses a key limitation of existing LSBD methods that assume known group structure, and the work constitutes a valuable contribution to unsupervised representation learning.

**Strengths:**

1. This paper proposes a complete framework for automatically discovering group structure and utilizing it for disentanglement, addressing the limitation of prior LSBD methods that rely on known group structure. This contribution demonstrates considerable novelty and significance.
2. This paper provides solid theoretical foundations for the proposed method while conducting systematic evaluation on multiple datasets, comparing not only standard disentanglement metrics but also evaluating long-term prediction and generalization capabilities.

**Weaknesses:**

1. The treatment of non-symmetry-based disentanglement methods is insufficient. While the paper situates itself within the LSBD literature, it provides minimal discussion of how symmetry-based approaches compare to traditional disentanglement methods.
2. The experimental comparison is confined to symmetry-based methods, leaving the performance of the proposed approach against mainstream baselines such as QLAE and FactorQVAE unclear.
3. The extensive mathematical notation throughout the manuscript hinders readability. A consolidated symbol table would significantly improve accessibility for readers less familiar with group theory.

**Questions:**

1. Could the authors elaborate on the tangible advantages of symmetry-based methods, particularly when augmented by the improvements proposed in this paper, over mainstream approaches like QLAE and FactorQVAE?
2. How realistic are the underlying assumptions in practical settings? All experiments employ relatively simple synthetic datasets. Would the assumptions hold on more complex benchmarks such as MPI3D or Isaac3D?

---

> ### Author Response · Authors · 2025-11-21
>
> We thank you for your valuable feedback on our submission, we adress each point blow.
>
> ### Weakness 1, Weakness 2 and Question 1
> Traditional disentanglement methods (β-VAE, Factor-VAE, DIP-VAE I/II), called _unsupervised methods_ in our paper, are reported in Appendix H.4. It is not part of the main text as they perform significantly worse than LSBD methods. We forgot to explictly mention those experiments in the main text, we added a sentence in section 5.3 to correct our mistake. More recent traditional disentanglement methods such as FactorQVAE could perform better than the traditional disentanglement methods used in our paper but we expect them to remain weaker than LSBD methods since they rely on less information.
>
> ### Weakness 3
> We added a symbol table at the end of Appendix A containing the most important symbols. Please let us know if you find omissions.
>
> ### Question 2
> To have theoritical guarantees, the assumptions are quiet restrictive coming from the LSBD framework itself (group properties, independant dynamics) and from our assumptions (disentangled action set, injective generative function),  it would be a next step to relax some assumptions. However the method may still work empirically without some of these assumptions, though without guarantees.
>
> But note that we added a more realistic experiment in section 5.6 with realistic images of MPI3D and continuous symmetry group.

---

> > ### Comment · Reviewer_GrsJ · 2025-11-27
> >
> > Thank you for the detailed rebuttal. I have no further comments.

---

### Official Review · Reviewer_7wnh · 2025-10-31

**Soundness:** 2
**Presentation:** 3
**Contribution:** 2
**Rating:** 4
**Confidence:** 3

**Summary:**

This paper tackles unsupervised symmetry group discovery for Linear Symmetry-Based Disentanglement (LSBD). It proposes a two-stage pipeline: (i) learn an “entangled” action representation and encoder via an Action-VAE trained on transitions, then (ii) cluster actions into subgroup factors using a group-theoretic pseudo-distance, and finally (iii) learn a block-masked GMA-VAE that enforces LSBD without assuming subgroup structure. The paper proves identifiability under injectivity and “disentangled action set” assumptions and evaluates on Flatland-style grids with color cycles/permutations, COIL rotations with permutations, and 3DShapes. Reported wins are on LSBD metrics and long-horizon prediction versus Forward-VAE/SOBDRL/LSBD-VAE variants.

**Strengths:**

1. Clear LSBD pipeline that separates (i) equivariant pretraining, (ii) group discovery, (iii) block-structured LSBD learning, with explicit assumptions and a clustering rule grounded in group algebra.

2. Good baseline experiments within the LSBD family (Forward-VAE/SOBDRL/LSBD-VAE variants) and consistent reporting on multiple disentanglement metrics and multi-step prediction.

3. Reproducibility: code, dataset generation, and hyperparameters are described and (per the authors) released.

**Weaknesses:**

1. **Lack of realistic interactive experiments.**
The environments are synthetic (Flatland, COIL with permutations, 3DShapes). There are no tests on widely used embodied/control suites (e.g., DeepMind Control/ProcGen/Habitat/ManiSkill) where continuous groups (SO(2), SE(2), SE(3)) and sensor noise dominate, and where symmetries are only approximate. By contrast, prior interactive symmetry/LSBD works motivate interaction explicitly and evaluate on non-trivial dynamics (e.g., SOBDRL/Forward-VAE), and the broader equivariant RL literature demonstrates robustness/sample-efficiency gains in control domains—precisely the use-case this paper claims to address.

2. **Strong assumptions.**
Identifiability and clustering rely on (i) fully injective observations, (ii) a disentangled action set with each action belonging to exactly one subgroup, and (iii) the existence of short compositions mediating within-subgroup relations. These are rarely true in practical agents (redundant actuators, coupled controls, unmodelled dynamics). No robustness analyses are provided for violations (e.g., aliased actions, missing transitions, approximate commutativity). The empirical advantage may be specific to the LSBD formulation and the chosen metrics.

3. **Limited group diversity and no continuous-group evaluation.**
All experiments use finite groups (cyclic, symmetric). There is no assessment on continuous groups or approximate symmetries (e.g., SE(2)/SE(3) rotations and translations, scaling), where many RL/control tasks live and where identifiability and clustering are harder.

4. **Limited discussion of related work.** This paper is missing a lot of citations and related works from the group structured representation learning in interactive environments literature [1, 2]. Many of these work uses actions of the environment to disentangle or induce structure in the latent space and shows it's benefits in reinforcement learning etc. I would suggest the authors to do a more thorough literature review.

[1] Learning Symmetric Embeddings for Equivariant World Models. JY Park et. al.
[2] Structuring Representations Using Group Invariants. M Shakerinava et. al.

**Questions:**

1. How does action clustering perform when the action set is not strictly disentangled (shared actuation across factors), or when a subset of transitions is missing/noisy? Please report clustering accuracy and downstream LSBD under controlled violations.

2. Can the pseudo-distance and masking be extended to continuous Lie groups (SO(2), SE(2)/SE(3))? If not, what fails? Compare against Lie-symmetry discovery approaches.

3. Can you demonstrate that discovered groups help in standard control tasks by instantiating ρ inside an equivariant policy/value network (as in group-equivariant RL)? Measure sample efficiency and generalization versus non-equivariant baselines.

4. Partial observability: Your proofs assume injective observations. What happens in POMDPs (occlusions, distractors)? Any way to incorporate history (RNN/state-space models) while preserving identifiability?

---

> ### Author Response · Authors · 2025-11-21
>
> We thank you for your valuable feedback on our submission. We would like to clarify that the goal of the paper is not to address a specific reinforcement learning use case, but rather to propose and analyze a method for unsupervised disentangled representation learning that does not target any particular downstream task. We understand that disentangled representations are often motivated by potential applications in reinforcement learning. However, our paper does not make claims regarding RL performance, nor does it rely on RL as a use-case. Nevertheless, since the framework used is formulated around an action–environment interaction model, we have good reasons to believe that our results could also carry over to reinforcement learning settings, even though we do not investigate this direction in the present work.
>
> ### Weakness 1
> We agree that evaluating the method on more realistic scenarios such as DeepMind control would be valuable. Our focus in this paper is on understanding the core properties of the proposed approach in controlled settings, but extending the evaluation to more realistic environments is indeed a natural next step. We have clarified this in the conclusion and plan to address it in future work.
>
> ### Weakness 2
> We agree that our three assumptions are strong and limit the scope of applicability. However, as argued in the paper, they enable theoretical guarantees of disentanglement.
>
> Regarding robustness, Section 5.2 evaluates action clustering under missing transitions and shows that the correct action partition is still recovered. Similarly, the generalization task in Section 5.5 demonstrates that GMA-VAE is robust to missing transitions. We additionally added disentanglement results under missing transitions in Appendix H.4 in Tables 11 to 14, showing that the disentanglement metric remains unaffected by the missing transition.
>
> ### Weakness 3
> While this paper focuses on finite action groups, we fully agree on the importance of continuous groups in practical applications. We therefore added an experiment with the continuous group $G=SO(2)\times SO(2)$ on the MPI3D dataset (Section 5.6). As noted there, the action clustering method cannot be applied to such groups, but GMA-VAE can, and its performance is comparable to LSBD state-of-the-art methods.
>
> ### Weakness 4
> We did not include such literature because the paper focuses on disentanglement rather than group-equivariant networks or RL. Nevertheless, we agree that our method is related to group-equivariant networks and we will add relevant references in the Section 6 in the following days.

---

> ### Author Response · Authors · 2025-11-21
>
> ### Question 1
> If the actions are not strictly disentangled there are two cases:
>
> (a) When the action set satisfies Assumption 2 and 3 after merging specific subgroups, for instance $G=G_1\times G_2\times G_3$ with $\mathcal G=(G_1\times G_2)\cup G_3$ satisfies Assumption 2 and 3 after merging $G_1$ and $G_2$. In such case, the clustering method will merge $G_1$ and $G_2$. The resulting representation learned by GMA-VAE will be only partially disentangled but the equivariance will still hold.
>
> (b) When the action set cannot satisfy Assumption 2 and 3 even after merging subgroups, for instance there is an action $g$ that cannot be composed to retrieve any other action, then this action will be clustered alone. The resulting representation learned by GMA-VAE will not satisfy the equivariance as $g$ will only affect its own subvector space despite being in the same subgroup than other actions.
>
> Regarding missing transitions, Section 5.2 already shows that the clustering method still recovers the correct decomposition in this setting. For noisy data, we are confident that neural networks can absorb moderate observational noise, the effect of noise on transitions is less clear and will be investigted in futur works.
>
> ### Question 2
> The pseudo-distance extends to continuous Lie groups as it doesn't rely on any assumption on $G$ or $\mathcal G$. However the clustering algorithm of Step 2 cannot be applied on Lie groups as it relies on a finite $\mathcal G$.
>
> If the Assumptions 1 and 2 are satisfied and the action clustering is given as prior knowledge, then the masking method can still be used. In such setting, we evaluate GMA-VAE and compare it to HAE and SOBDRL in section 5.6.
>
> ### Question 3
> Reinforcement learning applications could serve as an interesting downstream task for evaluating our method, but they fall outside the scope of this paper. Nevertheless, we assess generalization in a prediction task in Section 5.5, and as discussed above, we have good reasons to believe that this form of generalization may transfer effectively to RL settings.
>
> ### Question 4
> In the POMDP case, all proofs break down as highlighted by Theorem 1. Adding history to the state is insufficient, since it breaks the group structure: applying an action then its inverse yields the same observation but not the same history. There may be more sophisticated ways to recover theoretical guarantees, but to the best of our knowledge no simple solution exists. The method may still work empirically, though without guarantees.

---

> ### Author Response · Authors · 2025-11-27
>
> We have added a paragraph on group-equivariant neural networks in Section 6. In addition, we introduced a new experiment with noisy actions in Section 5.6 in order to provide a more realistic evaluation setting. In this setup, we compare GMA-VAE to related LSBD methods and show that it exhibits greater resilience to noise. We hope that these additions address some of the concerns raised in Weakness 1 and 2.

---

### Official Review · Reviewer_B8Yx · 2025-11-01

**Soundness:** 4
**Presentation:** 3
**Contribution:** 4
**Rating:** 8
**Confidence:** 2

**Summary:**

They propose an algorithm to learn and disentangle the group structure underlying an environment, through interactions with this environment. Unlike previous methods, their method does not assume specific subgoup properties.

**Strengths:**

My understanding of the framework is relatively superficial and I did not check the maths carefully.
Strengths:
- Careful explanations.
- Thorough comparison with other methods and with different metrics of disentanglement, which show an advantage over other unsupervised methods.

**Weaknesses:**

- How is the method less supervised than LSBD-VAE? It seems pretty supervised to me, with access to actions and consequences of these actions? What does it mean that "Both of our methods rely on a strong assumption which requires the available actions to be disentangled"?

**Questions:**

- The "Geomancer" should probably be cited as they somehow address a related question: https://arxiv.org/abs/2006.12982
- How to understand the problem of separating subgroups in different dimensions, when it is not always possible with a continuous encoder, as shown in https://arxiv.org/abs/2102.05623?
- How did Caselles-Dupre ́et al. (2019) demonstrate that symmetry-based disentanglement is only possible when access is granted to transitions? Consider explaining this in the paper.
- "Such an approach to disentanglement," => these alternative frameworks seem only tangentially relevant, and could be moved to appendix.

---

> ### Author Response · Authors · 2025-11-21
>
> We thank you for your valuable feedback on our submission, we adress each point blow.
> ### Weakness
> We agree that the terminology _supervised_, _self-supervised_, and _unsupervised_ may be confusing since we use it in a nonstandard sense. As detailed in section 5.1, we refer to a method as _supervised_ when the action representation $\rho$ is provided as prior knowledge while the _self-supervised_ methods require to learn $\rho$. Under this terminology, LSBD-VAE is the only supervised method.
>
> The sentence "Both of our methods rely on a strong assumption which requires the available actions to be disentangled" refers to Assumption 2 which requires every action to affect only one generative factor. Among the three assumptions, we consider this one the most restrictive, yet it is required both for action clustering and for disentanglement.
> Note that most of the works in the field are actually relying on the very same assumptions, maybe not in such an explicit way, see Appendix H.2.
>
> ### Question 1
> We have added a sentence about unsupervised LSBD methods including this article in the related work section.
>
> ### Question 2
> Thank you for bringing this reference [1] to our attention as we were not aware of this work. While it indeed establishes the impossibility of disentanglement in the LSBD framework, we would like to emphasize that the result relies on two assumptions that do not hold in our setting.
>
> The main assumption of the cited work is that the world set is continuous while the action group is finite. This covers a very specific case. In all our experiments, either both the world set and action set are continuous, or both are finite. Moreover, their result does not invalidate Theorem 3 in our paper: Theorem 3 requires the loss to converge to its minimum of 0, which seems not possible in the specific case treated by the cited work.
>
> Another assumption made in [1] is the fact that LSBD representation must necessarily encode the invariant and variant features in two distinct spaces. Here we aim to show that it is not required in our formulation of the LSBD framework. That's because the disentanglement of the LSBD framework is only defined through actions, if no action modifies a feature, then this feature does not need to be disentangled. Moreover in the formulation of [1], it seems implicit that the disentanglement is necessary with respect to $\prod_k G_k$, whereas we allow the disentanglement with respect to subgroups such as $\prod_k\langle \mathcal G_k\rangle$.
>
> To illustrate this, consider an environment where we observe a rotating colored arrow, $\mathcal W=\textbraceleft (c,\theta)\textbraceright$ with $c\in\textbraceleft1,2\textbraceright$ the arrow color and $\theta\in\textbraceleft0,\pi/2,-\pi/2,\pi\textbraceright$ the orientation. Suppose the color is invariant meaning that there is no action to change the color ie $\mathcal G_C\subset\textbraceleft e_C\textbraceright$ and $\theta$ is a variant feature meaning that $\mathcal G_\theta\not\subset\textbraceleft e_\theta\textbraceright$. [1] states that a LSBD representation should necessary encode $c$ and $\theta$ into two different subspaces, however the representation $f:(c,\theta)\mapsto ce^{i\theta}\in\mathbb R^2$ is LSBD with respect to $\langle\mathcal W, b,\prod_k\langle \mathcal G_k\rangle\rangle$. Indeed let's take $\mathcal Z_\theta=\mathbb R^2$ and $\mathcal Z_C=\textbraceleft(0,0)\textbraceright$ we therefore have $\mathbb R^2=\mathcal Z_\theta\oplus\mathcal Z_C$. Then for any $g\in\langle \mathcal G\rangle$ we have $g=(g_\theta, e_C)$ and therefore for $w\in\mathcal W$, $g\cdot_{\mathcal Z}f(w)=(g_\theta\cdot f(w), e_C\cdot 0_{\mathbb R^2})$ which implies disentanglement according to Definition 1.
>
> Do you think that those remarks should be added to the paper ?
>
> [1] D. Bouchacourt, M. Ibrahim, S. Deny, Addressing the Topological Defects of Disentanglement via Distributed Operators, NeurIPS 2021
>
> ### Question 3
> We added a brief proof sketch and included the precise theorem from the cited article at the end of Section 2.
>
> ### Question 4
> We removed this part

---

> > ### Comment · Reviewer_B8Yx · 2025-11-25
> >
> > I am satisfied with the answer. The authors could include the discussion on [1] as they see fit.

---

### Official Review · Reviewer_9xAV · 2025-11-01

**Soundness:** 3
**Presentation:** 2
**Contribution:** 3
**Rating:** 6
**Confidence:** 3

**Summary:**

This paper studied symmetry-based disentanglement learning. To remove assumption about the symmetry group’s structure,  this work introduces a novel, multi-step approach to learning disentangled representations by having an agent autonomously discover the underlying symmetry group structure of its environment. The proposed method consists of three main steps: 1) Learn an Entangled Representation (A-VAE). 2) Discover the Group Structure. 3) Learn a Disentangled Representation (GMA-VAE). The authors validate their method on three environments with different group structures (Flatland, COIL, and 3DShapes). The results show that their action clustering algorithm perfectly recovers the ground-truth group decomposition , and the final GMA-VAE model achieves disentanglement performance comparable to supervised methods that are given the group structure in advance, outperforming other self-supervised approaches.

**Strengths:**

1. **Novelty:** The core contribution—learning an LSBD representation via three steps—is instructive for disentanglement learning. This shifts the paradigm from requiring prior knowledge to autonomously learning it from interaction.
2. **Theoretical Grounding:** The paper is built on a solid theoretical foundation, providing formal proofs for its key claims. This guarantees the existence of actions belonging to the same subgroup (Theorem 2) and the  of learning a Linear Symmetry-Based Disentangled (LSBD) representation (Theorem 3).  This level of rigor adds significant weight to the proposed methods.
3. **Strong Empirical Validation:** The method is thoroughly tested across multiple benchmarks. The final disentanglement performance is shown to be on par with a supervised method (LSBD-VAE), which is a very strong result.
4. **Other properties:** The paper goes beyond standard disentanglement metrics to show the other properties of the learned representations. The experiments on long-term prediction clearly demonstrate that the entangled methods prefer short-term predictions, while self-supervised ones prefer long-term predicitons.

**Weaknesses:**

1. Multi-Stage Pipeline: The method is not end-to-end. It requires training two separate models sequentially: first the A-VAE to learn action matrices, and then the GMA-VAE to learn the final representation. The authors acknowledge this limitation, noting that a future direction would be to unify these steps into a single optimization process.
2. Limited Scope of Environments: The experiments are conducted on synthetic, visually simple datasets.  While these are well-suited for proving the group-theoretic concepts, it is unclear how the approach would scale to more complex, realistic, or stochastic environments where the underlying symmetries might be less perfect or harder to learn from raw pixels.
3. No visualisation evaluation for  reconstruction quality and latent traversal to explicitly demonstrate the tradeoff.
4. The number of MIG on 3Dshapes is low.

**Questions:**

Why the MIG score is low on 3Dshapes? The discussion about failing cases is important to understand the drawbacks of proposed method.

---

> ### Author Response · Authors · 2025-11-21
>
> We thank you for your valuable feedback on our submission, we address each point blow.
>
> ### Weakness 1
> Thank you for your comment. You are absolutely right to point out that our approach is based on two successive learning steps, a limitation that we have discussed, as you noted. At this stage, we do not plan to address this limitation in this submission, but it is of course an area for improvement that we are working on.
>
> ### Weakness 2
> We agree that our experimental settings rely on idealized environments. More complex experiments with noisy observations and/or noisy transitions could certainly be performed. While we are confident that neural networks can absorb moderate realistic observational noise, the effect of noise on transitions is less clear and will be investigted in futur works.
>
> As an attempt to include more realistic scenarios, and in the line of recommandations by Reviewer B8Yx, we have added new results with the MPI3D dataset (Section 5.6) which includes semi-realistic environments with continuous actions.
>
> ### Weakness 3
> We have added latent manipulation experiments in Appendix H.3, including reconstruction quality and latent traversals for GMA-VAE in both the discrete-action and newly added continuous-action settings.
>
> However, it is unclear to us what specific “tradeoff” you are referring to. If the intended tradeoff is the classical reconstruction–disentanglement tradeoff observed in methods such as β-VAE, we note that LSBD methods (including ours) do not suffer from this issue. They can simultaneously minimize reconstruction error and the disentanglement criterion.
>
> ### Weakness 4 and Question
> For each factor, the MIG metric measures the gap in mutual information between the two latent dimensions that are most informative about that factor. As a consequence, achieving a high MIG score requires each factor to be encoded in exactly one latent dimension. However, as discussed in Appendix E, most groups must be represented in at least two dimensions for the latent transition function to remain linear. For example, any subgroup $G\subset SO(2)$ requires a two-dimensional representation whenever $|G|>2$. In our datasets, the only exception is the subgroup $\mathfrak S_2$ in COIL2, which can indeed be represented in a single dimension with GMA-VAE.
>
> This specificity about the MIG metric which explains the low score values is explicitly mentioned in the text, see paragraph 5.3.

---

> > ### Author Response · Authors · 2025-11-27
> >
> > We introduced a new experiment with noisy actions in Section 5.6 in order to provide a more realistic evaluation setting. In this setup, we compare GMA-VAE to related LSBD methods and show that it exhibits greater resilience to noise. We hope that these additions address some of the concerns raised in Weakness 2.

---

> > ### Comment · Reviewer_9xAV · 2025-11-28
> >
> > thank you. I have no other concern.

---

### Author Response · Authors · 2025-12-03

We thank all reviewers for their constructive feedback. In response, we have made the following revisions to the paper and the provided code:

- **More realistic environment (9xAV, 7wnh, GrsJ).** We added Section 5.6, which introduces experiments under continuous symmetry groups (as suggested by reviewer 7wnh), noisy actions (as suggested by reviewers 9xAV and 7wnh), and on the MPI3D dataset containing realistic observations (as suggested by reviewer GrsJ). These experiments are designed to evaluate the disentanglement capabilities of our method in more realistic settings. The results show that our method performs comparably to state-of-the-art LSBD methods while exhibiting greater robustness to noise.
- **Robustness of the method (7wnh).** In addition to the action-clustering experiments under missing transitions, we have added disentanglement metrics evaluated in both missing-transition and noisy-transition settings. All the detailed results, reported in Appendix H.4, show that our method exhibits strong resilience to both types of perturbations.
- **Latent traversal and reconstruction quality (9xAV).** We added Appendix H.3, which presents latent manipulation experiments, including latent traversals and reconstruction quality, for both continuous and finite symmetry groups.
- **Related work (B8Yx, 7wnh).** We added a third paragraph to Section 6 providing a discussion of related work on equivariant neural networks as suggested by reviewer 7wnh. We also added a part about unsupervised LSBD methods in Section 3 as suggested by reviewer B8Yx.
- **Comparison with traditional methods (GrsJ).** We clarified that LSBD-based approaches outperform traditional disentanglement methods relying solely on observations. We now explicitly point readers to Appendix H.4, where detailed experimental results are provided.
- **Minor editorial improvements.** We made several refinements throughout the text and added additional implementation details in Appendix I.
- **Theoritical clarification.** We realized that Theorem 3 (and consequently Theorem 3′) does not require Assumption 3. We have therefore updated the statement of the theorem accordingly.

---

### Meta-Review · Area_Chair_Nzg3 · 2026-01-02

**Summary:**

The paper presents a symmetry-based disentanglement method that automatically discovers the underlying symmetry group without prior assumptions and learns high-quality disentangled representations. In response to the reviews, the authors added more realistic and robust experiments, improved qualitative analyses, and expanded related work and comparisons. Reviewers' major concerns have been fixed.

**Reviewer Concerns:**

Reviewers 9xAV, B8Yx, and GrsJ have confirmed in their post-rebuttal responses that their concerns have been addressed. Reviewer 7wnh did not post a post-rebuttal response; however, after reading the authors’ rebuttal, I believe their major concerns have been addressed as well.

**Reviewer Scores:**

Reviewer 7wnh may increase their score accordingly if they read the authors’ response and participate in the discussion.

---

### Decision · Program_Chairs · 2026-01-26

Accept (Poster)